# Conserved transcription factors promote cell fate stability and restrict reprogramming potential in differentiated cells

Maria A. Missinato[1,6], Sean Murphy[2,6], Michaela Lynott [1,6], Michael S. Yu[1], Anaïs Kervadec[1], Yu-Ling Chang[1], Suraj Kannan[2], Mafalda Loreti[1], Christopher Lee [1], Prashila Amatya[1], Hiroshi Tanaka[1], Chun-Teng Huang[3], Pier Lorenzo Puri [1], Chulan Kwon [2], Peter D. Adams [4], Li Qian [5], Alessandra Sacco[1], Peter Andersen [2] ✉ & Alexandre R. Colas [1] ✉

Defining the mechanisms safeguarding cell fate identity in differentiated cells is crucial to improve 1) - our understanding of how differentiation is maintained in healthy tissues or altered in a disease state, and 2) - our ability to use cell fate reprogramming for regenerative purposes. Here, using a genome-wide transcription factor screen followed by validation steps in a variety of reprogramming assays (cardiac, neural and iPSC in fibroblasts and endothelial cells), we identified a set of four transcription factors (ATF7IP, JUNB, SP7, and ZNF207 [AJSZ]) that robustly opposes cell fate reprogramming in both lineage and cell type independent manners. Mechanistically, our integrated multi-omics approach (ChIP, ATAC and RNA-seq) revealed that AJSZ oppose cell fate reprogramming by 1) - maintaining chromatin enriched for reprogramming TF motifs in a closed state and 2) - downregulating genes required for reprogramming. Finally, KD of AJSZ in combination with MGT overexpression, significantly reduced scar size and improved heart function by 50%, as compared to MGT alone post-myocardial infarction. Collectively, our study suggests that inhibition of barrier to reprogramming mechanisms represents a promising therapeutic avenue to improve adult organ function post-injury.

Cell fate identity is acquired during the process of differentiation and consists of the lineage-specific establishment of chromatin organization and gene expression, that support the function of specialized cells[1]. Although observed to be stable in differentiated cells, groundbreaking studies[2–4] have revealed that developmentally-inherited fate identity can be reprogrammed by forced expression of lineage-specific combinations of transcription factors (TFs) (reviewed in ref. [5]). In this context, a diversity of cell types including fibroblasts or endothelial cells, could be reprogrammed into induced pluripotent stem cells[4,6], neurons[7] or cardiomyocytes[8]. However, in most cases, only a small fraction of fate-challenged cells underwent reprogramming (reviewed in refs. [9,10]), thereby suggesting the existence of robust mechanisms opposing this process.

[1]Development, Aging and Regeneration Program, Sanford Burnham Prebys Medical Discovery Institute, 10901 North Torrey Pines Road, La Jolla, CA 92037, USA. [2]Division of Cardiology, Department of Medicine, Johns Hopkins University School of Medicine, Baltimore, MD 21205, USA. [3]Viral Vector Core Facility Sanford Burnham Prebys Medical Discovery Institute, 10901 North Torrey Pines Road, La Jolla, CA 92037, USA. [4]Tumor Initiation and Maintenance Program, Sanford Burnham Prebys Medical Discovery Institute, 10901 North Torrey Pines Road, La Jolla, CA 92037, USA. [5]McAllister Heart Institute, University of North Carolina at Chapel Hill, Chapel Hill, NC 27599, USA. [6]These authors contributed equally: Maria A. Missinato, Sean Murphy, Michaela Lynott. ✉e-mail: pander31@jhmi.edu; acolas@sbpdiscovery.org

Intense research during the past decade (reviewed in refs. [5,11]) has established that a rate-limiting step for cell fate conversion resides in the ability of reprogramming TFs to efficiently bind to their target DNA and activate destination cell type gene expression[12]. Consistent with this reprogramming paradigm, mediators of heterochromatin formation, such as histone chaperones[13], enzymes involved in histone H3[14,15], or DNA[16,17] methylation, were found to oppose cell fate reprogramming. In addition to chromatin-associated mechanisms, regulators of source and destination cell type transcriptomes such as TGFβ and inflammatory signaling pathways[18,19] or genes mediating RNA methylation[20,21], alternative polyadenylation[22], and splicing[23], were also found to limit cells' ability to reprogram. Collectively, these studies demonstrate that cell fate stability is regulated by the concomitant maintenance of cell type-appropriate chromatin architecture and transcription. In this context, we hypothesize the existence of mechanism(s) simultaneously controlling both regulatory dimensions to promote cell fate stability and oppose reprogramming in differentiated cells.

TFs are essential determinants of differentiation[24,25] and mediate their role via direct DNA binding to regulate both transcription and chromatin accessibility[26,27], and thus represent a biochemically-relevant class of proteins to mediate cell fate stability in differentiated cells. Consistent with our hypothesis, recent work from Gurdon and colleagues[28], has shown that long-term DNA association of lineage-specific TF, ASCL1, contributes to maintain gene expression and stabilize fate commitment in differentiating cells. Moreover, several studies have also shown that various TFs, including SNAI1[29], cJun[30], and Bright/ARID3A[31], oppose cell fate reprogramming by enhancing source cell type gene expression or repressing reprogramming TF-associated gene expression. However, a systematic evaluation of TFs as fate stabilizers has not been conducted to date, and in this context, we will explore whether such factors might be evolutionarily conserved and play lineage and/or cell type-specific or independent roles. Mechanistically, we will evaluate how fate-stabilizing TFs binding to the DNA might contribute to regulate chromatin accessibility dynamics and transcription to oppose the lineage reprogramming process. Finally, we will test whether the targeted inhibition of such fate stabilizers might enhance our ability to use reprogramming TFs to improve adult organ function post-injury.

To uncover fate stabilizers, we employed a genome-wide TF siRNA screen in a cardiac reprogramming (CR) assay using mouse embryonic fibroblasts, followed by the combinatorial evaluation of top-performing hits, and validated our findings in human primary cells (human fibroblasts and adult endothelial cells) in reprogramming assays probing three different lineages (cardiac, neural, iPSC). This approach led us to identify a conserved set of 4 TFs (ATF7IP, JUNB, SP7, and ZNF207 [AJSZ]) that robustly opposes cell fate reprogramming, as demonstrated by up to a six-fold increase in efficiency upon AJSZ knockdown in both lineage- and cell type-independent manners. Mechanistically, ChIP- and single-cell ATAC-seq analyses, revealed that AJSZ binds to AP-1 and STAT4/5/6 motif-enriched regions in open and closed chromatin, thereby limiting reprogramming TFs to access their target DNA and remodel chromatin. In parallel, ChIP- and RNA-seq data integration, followed by systematic gene testing, uncovered that AJSZ also promotes cell fate stability by downregulating the expression of a conserved set of genes, limiting cells' ability to undergo large-scale phenotypic changes. Finally, KD of AJSZ, in combination with MGT overexpression, reduced scar size and improved heart function by 100% as compared to no treatment and 50% as compared to MGT alone, 1 month after myocardial infarction. Collectively, this study uncovers (1) a mechanism by which conserved TFs promote cell fate stability in differentiated cells and (2) a promising target space to improve adult organ repair post-injury.

## Results

### Genome-wide TF screen identifies regulators of cell fate stability in mouse fibroblasts

To identify cell fate stabilizers, we employed immortalized mouse embryonic fibroblasts that carry a cardiac-specific fluorescent reporter (*Myh6-eGFP*) and a doxycycline-inducible cassette enabling the overexpression of cardiac reprogramming (CR) factors *Mef2c*, *Gata4*, and *Tbx5* (= iMGT-MEFs) (Fig. 1a and ref. [32]). In this context, doxycycline treatment (1 μg/ml) typically induces about 6% of fate-challenged cells to initiate CR and express *Myh6-eGFP* by day 3 (Supplementary Fig. 1a), thus highlighting that under these control conditions, yet-to-be-identified barrier mechanisms prevent most cells from undergoing cell fate reprogramming.

To determine whether TFs may play a role as fate stabilizers, we transfected iMGT-MEFs with a library of siRNAs directed against 1435 TFs, one day prior to the doxycycline treatment and quantified CR efficiency (= percentage of *Myh6-eGFP* + cells) by day 3, using a previously established automated and high-throughput fluorescence imaging approach[33,34]. In total, the screen identified 69 siRNAs that significantly increased CR efficiency (>1.25-fold, $p < 0.05$) as compared to siControl (Fig. 1b and Supplementary Data 1). Next, the top 20 hits were selected for validation using a different set of siRNAs (see Methods) and tested for function in the iMGT-MEFs assay. In total, this approach identified eight siRNAs that robustly increased CR efficiency (1.2−3.8-fold, $p < 0.05$) as compared to siControl, (Fig. 1c, d and Supplementary Fig. 1b), thus uncovering a role for Atf7ip, Foxa1, Hexim2, Junb, Smarca5, Sox15, Sp7, and Zfp207 as barriers to CR in MEFs.

Next, to evaluate whether identified TFs functionally interact with each other to mediate their fate-stabilizing role, we assembled and tested all possible combinations of the eight validated siRNAs (255 combinations in total), in the iMGT-MEFs assay (Fig. 1e). In total, this approach identified four siRNA combinations that significantly enhanced CR efficiency beyond best single hit: siAtf7ip (Fig. 1f and Supplementary Data 2). Remarkably, the most efficient combination, consisting of siRNAs targeting Atf7ip, Junb, Sp7, and Zfp207 (Supplementary Fig. 2a–e), increased the percentage of *Myh6-eGFP* + cells from ~6 to ~36%, representing a ~6-fold increase in CR efficiency as compared to siControl and ~1.5-fold increase as compared to siAtf7ip alone (Fig. 1g, h). Taken together, our results identify Atf7ip, Junb, Sp7, and Zfp207 (hereafter referred to as AJSZ), which are members of the ATF-interacting, AP-1, SP, and Zinc Finger TF families, respectively, as functionally interacting barriers to reprogramming, mediating cell fate stability in MEFs.

### AJSZ mediates cell fate stability in human primary fibroblasts

To determine whether the fate-stabilizing function of AJSZ is conserved across species, we first confirmed their endogenous expression in human primary cells such as dermal fibroblasts (HDFs) (Fig. 2a). Next, we developed a human CR assay, where MGT is overexpressed by retrovirus delivery in HDFs. First, we determined the percentage of transduced cells by immunofluorescence and observed that ~25−30% of HDFs expressed MGT (Supplementary Fig. 3a, b). Transfection with siAJSZ led to a ~50% reduction of AJSZ protein levels by day 3 (Supplementary Fig. 3c, d). Remarkably, in this context, KD of AJSZ led to a significant increase of cardiac-specific gene expression (2−8-fold, $p < 0.05$), including transcripts associated with cardiac contractility (*ACTC1*, *MYL7,* and *TNNT2*), ion channels (*SCN5A* and *RYR2*), and cardiokines (*NPPA* and *NPPB*), as compared with siControl-transfected HDFs (Supplementary Fig. 4a). Consistent with these findings, immunostaining for cardiac marker α-actinin (ACTN2) by day 30 after MGT overexpression, revealed a ~3.2-fold increase in the generation of ACTN2 + cells (from 5 to 16%) in siAJSZ condition as compared to siControl ($p < 0.05$; Fig. 2b, c and Supplementary Fig. 4b). Importantly, the increase in CR efficiency observed after AJSZ KD was also accompanied by the acquisition of cardiac-

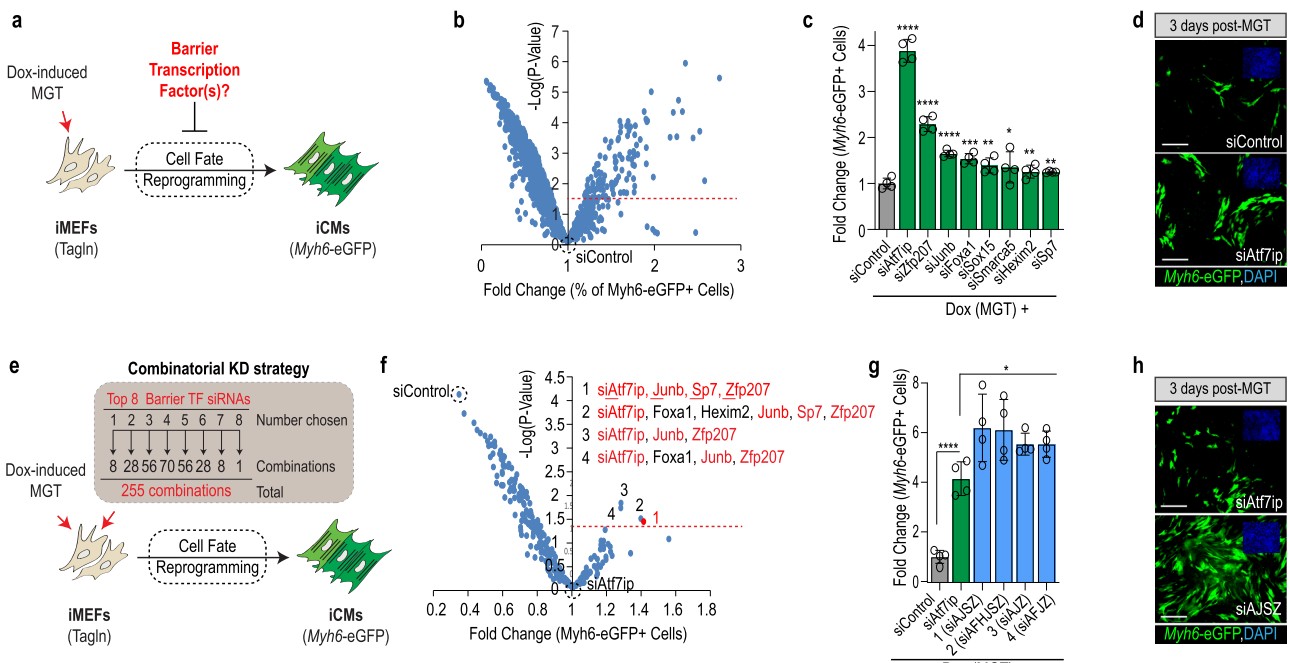

**Fig. 1 | Genome-wide TF screen identifies Atf7ip, JunB, Sp7, and Zfp207 (ZNF207) as barriers to cell fate reprogramming. a** Schematic of the cardiac reprogramming assay and experimental rationale. **b** Volcano plot depicting genome-wide TF screen results. X-axis shows % of Myh6-EGFP + positive normalized to siControl. Y axis represents −log of P value as compared to siControl. The screen was run in experimental quadruplicate. **c** Validation of top 20 hits identifies eight siRNAs with confirmed activity. Student's *t*-test: *$p < 0.05$, **$p < 0.01$, ***$p < 0.001$, and ****$p < 0.0001$. **d** Representative images of top hit siAtf7ip- and siControl-transfected iMGT-MEFs on day 3 after MGT induction. Myh6-eGFP is shown in green and nuclei are stained with DAPI (blue, top right insets). **e** Schematic of the combinatorial screening approach. A total of 255 combinations was tested, each in quadruplicate. **f** Volcano plot depicting genome-wide combinatorial screen results. Data were normalized and compared to the most potent single hit (siAtf7ip). 1–4 indicates top combinations that were significantly more potent than siAtf7ip alone ($p < 0.05$). siAtf7ip, siJunb, siSp7, and siZfp207 (siAJSZ) are shown in red. **g** Visualization of top 4 siRNA combinations in a histogram plot as in (**c**). **h** Representative images of iMGT-MEFs transfected with siAJSZ (siAtf7ip, siJunb, siSp7, and siZfp207) 3 days after MGT induction. *n* = 4 per condition for all data in this figure. Groups were compared using two-tailed unpaired analysis. Data in the figure are presented as mean values ± standard deviation. Scale bars: 50 μm. **a**, **e** Schematics are modified from Cunningham, T. J. et al. Id genes are essential for early heart formation. Genes & development, https://doi.org/10.1101/gad.300400. 117 (2017). - CC-BY 4.0.

specific structural and functional phenotypes, including sarcomeric-like structures and calcium handling activity (Supplementary Fig. 4c, d and Supplementary Movie 1). Collectively, these results indicate that the fate-stabilizing function of AJSZ is conserved between mouse and human fibroblasts.

### AJSZ regulates cell fate stability in a lineage- and cell type-independent manner

To assess whether the fate-stabilizing role of AJSZ is restricted to the cardiac lineage or whether these findings could be generalized to other lineages, we established a neural reprogramming assay by over-expressing, *Ascl1*, *Brn2*, and *Mytl1* (ABM)[35] in HDFs (Supplementary Fig. 5a and Methods). Remarkably, compared with siControl-transfected cells, AJSZ KD enhanced the generation of MAP2 + and TUJ1 + neuron-like cells by ~2.3-fold (from 7.3 to ~17%) at day 3 (Fig. 2d, e) and concomitantly increased expression of neuron-specific markers[36], including *vGLUT2*, *GAD67*, *PVALB*, and *SYN1* by ~2-fold ($p < 0.05$) (Supplementary Fig. 5b), thus suggesting that the AJSZ-mediated regulation of cell fate stability in HDFs is lineage-independent.

Next, to determine whether the role of AJSZ is restricted to direct reprogramming processes or whether it could be generalized to somatic reprogramming, we established an iPSCs reprogramming assay by overexpressing pluripotency TFs, *OCT4*, *KLF4*, *SOX2*, and *cMYC* (OKSM)[37] in HDFs (see Methods). Remarkably, AJSZ KD increased the generation of NANOG + cells (Fig. 2f) by ~2-fold (from 9 to ~18%) and expression of multiple pluripotency markers[37] at day 7, including *DPPA2*, *DPPA4*, ZFP42 (*REX1*), and *NANOG* by ~2–6-fold ($p < 0.05$)

(Supplementary Fig. 5c, d), as compared to siControl. In conclusion, our findings suggest that AJSZ generally restricts fibroblasts' ability to undergo cell fate reprogramming.

Finally, given the widespread expression of AJSZ in most human adult tissues (see https://www.proteinatlas.org/[38]), we asked whether their fate-stabilizing role would be conserved in non-fibroblast cell types. First, we confirmed AJSZ expression in an alternative cell type, such as human aortic endothelial cells (HAECs) and KD efficiency upon siAJSZ transfection (Supplementary Fig. 6a, b). Next, we developed a CR assay in HAECs (see Methods) and observed that AJSZ KD (Supplementary Fig. 6c, d) enhanced cardiac gene expression (2–10-fold, $p < 0.05$) on day 3 (Supplementary Fig. 6c, d). Consistent with these observations, AJSZ KD also led to a significant increase (~1.5-fold, $p < 0.05$) in the generation of ACTN2 + iCMs (Fig. 2h, i and Supplementary Fig. 6e) as compared siControl by day 20. In sum, our results demonstrate that AJSZ promotes cell fate stability and opposes cell fate reprogramming in both a lineage (cardiac, neural, and iPSC) and cell type (fibroblasts, endothelial cells)-independent manner.

### AJSZ-mediated fate-stabilizing mechanisms are actively deployed at the ground state in fibroblasts

To gain insight into AJSZ mode of action, we first determined whether the timing of AJSZ KD relative to reprogramming factors over-expression would influence reprogramming efficiency (Supplementary Fig. 7a). Strikingly, while siAJSZ transfection on day −1 relative to Dox-mediated MGT induction robustly enhanced CR efficiency, AJSZ KD at day +1 failed to enhance CR efficiency (Supplementary Fig. 7b, c).

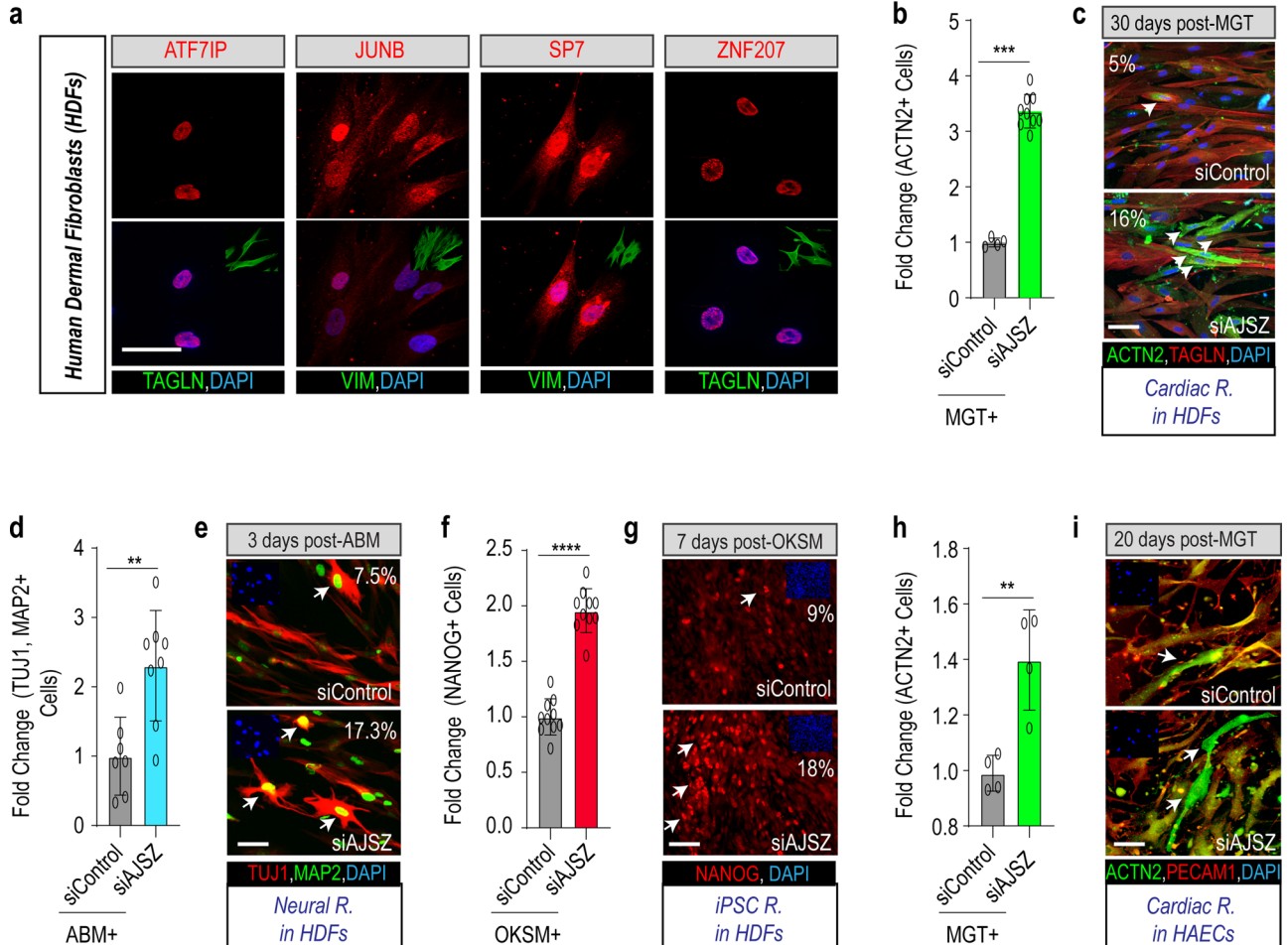

**Fig. 2 | Lineage- and cell type-independent role for AJSZ as barriers to reprogramming. a** Expression of AJSZ in HDFs. Immunostaining for AJSZ is shown in red and fibroblast markers Vimentin (VIM) or Transgelin (TAGLN) are shown in green. Nuclei are stained with DAPI. $n = 4$ per condition. **b** Quantification of the % of cardiac marker ACTN2-expressing cells normalized to MGT + siControl condition after 30 days of cardiac reprogramming in HDFs. $n = 5$ and $n = 9$ for siControl and siAJSZ treated, respectively. **c** Representative images of HDFs treated with MGT + siControl or MGT + siAJSZ and immunostained for cardiac (ACTN2, green) and fibroblast (TAGLN, red) markers. Nuclei are stained with DAPI (blue). White arrows indicate iCMs. **d** Quantification of the % of MAP2 (green) and TUJ1 (red)-double-positive cells, 3 days after overexpression of Ascl1, Brn2, and Mytl1 (ABM) in HDFs. $n = 7$ and $n = 8$ for siControl and siAJSZ treated, respectively. **e** Representative images of HDFs treated with ABM + siControl or ABM + siAJSZ and immunostained for MAP2 (green) and TUJ1 (red). Nuclei are stained with DAPI (blue, top left insets).

White arrows indicate TUJ1 + MAP2 + double-positive cells = neurons. **f** Quantification of the % of NANOG-positive cells, 7 days after overexpression of OCT4, KLF4, SOX2, and cMYC (OKSM) in HDFs. $n = 10$ per condition. **g** Representative images of HDFs treated with OKSM + siControl or OKSM + siAJSZ and immunostained for pluripotent marker NANOG (red). Nuclei are stained with DAPI (blue, top left insets). White arrows indicate NANOG-positive cells= iPSCs. **h** Quantification of the % of ACTN2 + 20 days after MGT overexpression. **i** Representative images of HAECs treated with MGT + siControl or MGT + siAJSZ and immunostained for the endothelial marker (PECAM1, red) and the cardiac marker (ACTN2, green). $n = 4$ per condition. Nuclei are stained with DAPI (blue). White arrows indicate iCMs. Scale bars: 50 μm. Student's $t$-test: *$p < 0.05$, **$p < 0.01$, ***$p < 0.001$, and ****$p < 0.0001$. Groups were compared using two-tailed unpaired analysis. Data in figure are presented as mean values ± standard deviation.

Interestingly, transfection of siAJSZ at day −2 or −3 progressively reduces AJSZ KD ability to enhance CR efficiency (Supplementary Fig. 7d, e). In this context, expression analysis by qPCR, reveals that transfection of siAJSZ at day-1 elicits a more efficient KD of the barrier factors than transfection of siAJSZ on day-3, at the time when MGT reaches a maximum of expression (day 1 and 2) and induce cardiac reprogramming (Supplementary Fig. 7f, g). Thus collectively, these results indicate that (1) cells' ability to reprogram is inversely proportional to barrier TF expression levels and (2) AJSZ-mediated fate-stabilizing mechanisms are actively deployed and maintained at ground state in fibroblasts, prior to the cell fate challenge.

## Motif and chromatin state-specific binding of AJSZ in unchallenged HDFs

To characterize the molecular nature of AJSZ-mediated fate-stabilizing mechanisms, we examined AJSZ interaction with the DNA in unchallenged HDFs using ChIP-seq (Fig. 3a). This approach identified 91,196 replicated peaks (binding sites) for JUNB; 44,100 for ATF7IP; 19,169 for SP7, and 4135 for ZNF207 across two duplicates (Fig. 3b and Supplementary Data 3), thereby revealing a pervasive association of the four TFs with the chromatin, where JUNB and ATF7IP contributed to >85% of binding events. Next, we generated a genome-wide chromatin accessibility profile of unchallenged HDFs using ATAC-seq and determined AJSZ binding distribution in regard to the chromatin state (open or closed). This analysis revealed that JUNB and ZNF207 preferentially bound to regions of open chromatin (78 and 94% of total binding, respectively), while ATF7IP and SP7 mainly interacted with the closed chromatin (79 and 97% of total binding, respectively) (Fig. 3c). In addition to these main interactions, 21% of ATF7IP (~9000 binding sites) and 22% of JUNB binding (~20,000 binding sites) occurred in open and closed chromatin respectively, thus revealing ATF7IP and JUNB ability to interact with both chromatin states albeit with

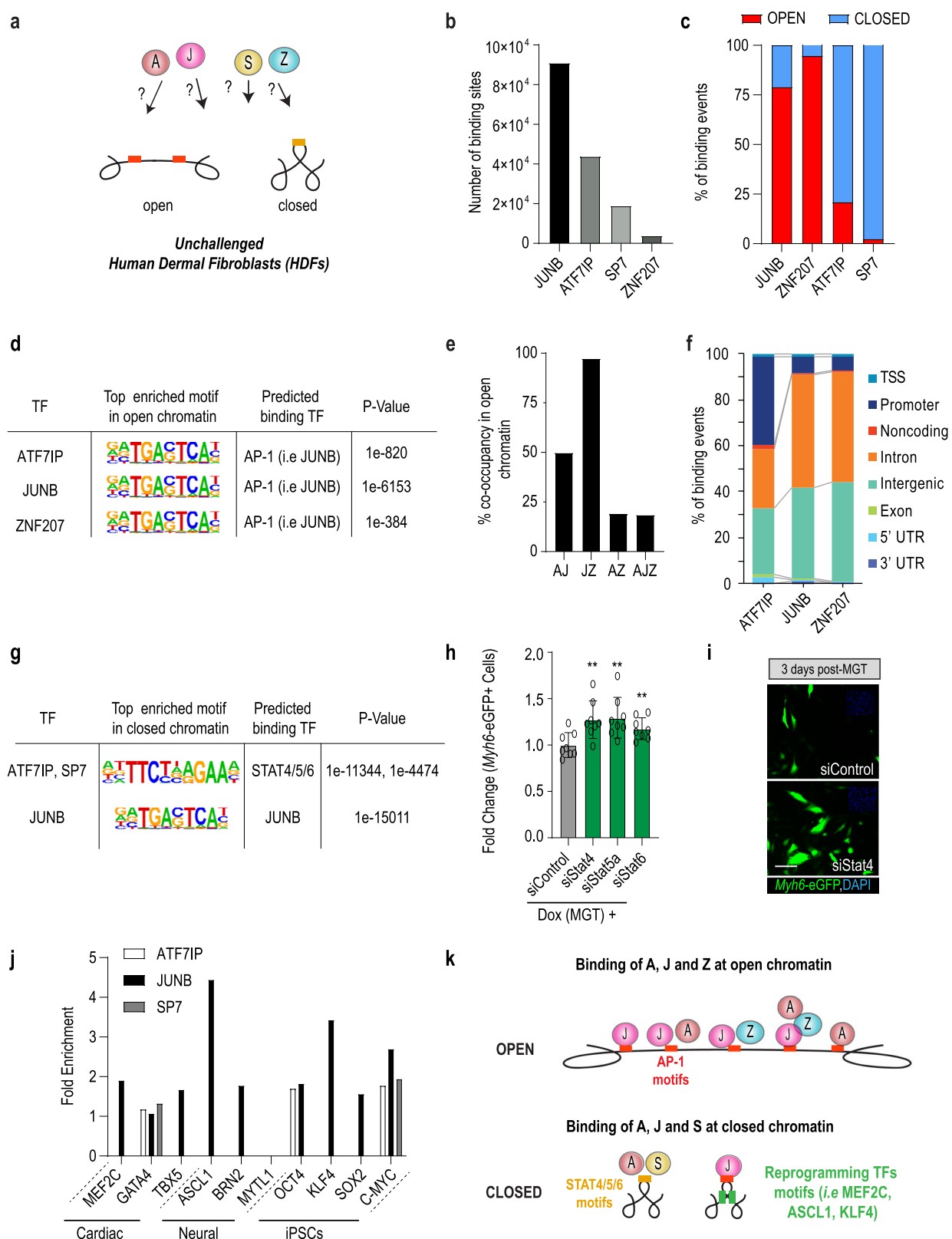

differential frequency. In sum, our data show that in unchallenged HDFs, the four fate-stabilizing TFs interact with the DNA in a regionalized manner, with ATF7IP, JUNB, and ZNF207 engaging the open chromatin and ATF7IP, JUNB, and SP7 the closed chromatin.

To delineate ATF7IP, JUNB, and ZNF207 binding properties at regions of open chromatin, we performed a motif enrichment analysis and identified prototypical AP-1 TF recognition elements (-TGACTCA-)[39],

including putative JUNB binding motif, as top enriched sequences for the three TFs (Fig. 3d and Supplementary Data 4). Consistent with these findings, the co-occupancy analysis revealed that 50% of ATF7IP and 97% of ZNF207 binding sites were co-occupied by JUNB (Fig. 3e and Supplementary Data 5). Moreover, immunoprecipitation of ZNF207 revealed a direct interaction with ATF7IP and JUNB in HDFs (Supplementary Fig. 8a), thus suggesting that JUNB might contribute to recruit

**Fig. 3 | AJSZ binding properties in open and closed chromatin in HDFs.**
**a** Chromatin binding properties of AJSZ in human dermal fibroblasts (HDFs) have not been characterized. **b** Number of binding sites determined by ChIP-seq for each factor. Peaks were merged from two samples. **c** Percentage of AJSZ binding sites located in open or closed chromatin. scATAC-seq was used to determine the state (open or closed) of the chromatin. **d** Top enriched motif in open chromatin bound by AJZ. **e** Co-occupancy analysis for AJZ at regions of open chromatin. **f** AJZ binding distribution at annotated regulatory regions in open chromatin. **g** Top enriched motifs in closed chromatin bound by AJS. **h** siStat4, 5a, or 6 respectively increase

reprogramming efficiency as compared to siControl in iMGT-MEF assay. $n = 8$ biologically independent experiments for siControl and siStat4, 5a or 6 treated. Student's $t$-test: **$p < 0.01$. Groups were compared using two-tailed unpaired. Data were presented as mean values ± standard deviation. **i** Representative images for siStat4 and siControl-transfected iMGT-MEFs 3 days post-Dox(MGT) treatment. Scale bars: 50 μm. **j** Motif analysis at regions of AJS-bound closed chromatin show enrichment for reprogramming TF motifs (i.e., MEF2C, ASCL1, and KLF4). **k** Schematic summarizing AJSZ binding properties at regions of open and closed chromatin in HDFs.

ATF7IP and ZNF207 at AP-1 motif-enriched open chromatin regions. Next, analysis of binding distribution revealed that 54–65% of interactions occurred at regions associated with the regulation of transcription (i.e., promoter/TSS, TSS, and introns) (Fig. 3f). In this context, GO term analysis of core promoter-bound genes revealed enrichment for ontologies involved in a wide variety of cellular functions including *ncRNA* export from the nucleus, regulation of deacetylase activity and regulation of signal transduction by *p53* class mediator (Supplementary Fig. 8b–e and Supplementary Data 6). Finally, we also noted that a significant portion (26–43%) of ATF7IP, JUNB, and ZNF207 binding occurred at intergenic regions, thus suggesting a potential role for these interactions in the maintenance of these regions in an open state. In summary, here we find that ATF7IP, JUNB, and ZNF207 interaction with the open chromatin is mainly mediated by AP-1 motif binding and might contribute to (1) the regulation of transcription and (2) the maintenance of bound regions in an open state.

At regions of closed chromatin, ATF7IP and SP7 binding sites were most enriched for STAT4/5/6 motifs, while in contrast, JUNB binding sites were most enriched for AP-1 TFs motifs as in the open chromatin (Fig. 3g and Supplementary Data 7). Consistent with these observations, the co-occupancy analysis revealed that 70% of SP7 binding sites were co-occupied by ATF7IP, while less than 2% of ATF7IP and SP7 binding sites were co-occupied by JUNB (Supplementary Data 8), thus indicating that ATF7IP and SP7 on the one hand and JUNB on the other, bind to distinct domains of closed chromatin. Next, to determine if ATF7IP and SP7 interaction at STAT4/5/6 motif-enriched regions may contribute to a barrier to the reprogramming role, we transfected siRNAs directed against Stat4/5a/6 in the iMGT-MEFs assay and observed a ~1.3-fold increase in reprogramming efficiency as compared to siControl (Fig. 3h, i), thus suggesting that ATF7IP and SP7 may cooperate with STAT4/5/6 at discrete regions of closed chromatin to stabilize cell fate. Finally, given that a rate-limiting step for cell fate conversion resides in the ability of reprogramming TFs to access their target DNA, we next asked whether ATF7IP, JUNB, and SP7-bound closed chromatin might be enriched for putative reprogramming TFs motifs. Remarkably, this analysis uncovered that JUNB-bound closed chromatin was significantly enriched for multiple reprogramming TF motifs including, cardiac (MEF2C 1.89-fold, $p$ value = 1e-172 and TBX5 1.66-fold, $p$ value = 1e-366), neural (ASCL1 4.43-fold, $p$ value = 1e-854), and pluripotency motifs (KLF4 3.42-fold, $p$ value = 1e-71; cMYC 2.68-fold, $p$ value = 1e-101) (Fig. 3j), thus indicating that JUNB might contribute to maintain these regions in a closed state, thereby potentially limiting reprogramming TFs to access their target DNA. Taken together, these results indicate that in unchallenged HDFs, AJSZ binds to the DNA in a motif (AP-1 and STAT4/5/6)- and chromatin state (open or closed)-specific manner, thereby contributing to the regulation of transcription and chromatin architecture (Fig. 3k).

## AJSZ-mediated regulation of chromatin accessibility during cell fate reprogramming

Our data above show that AJSZ extensively binds to both open and closed chromatin in unchallenged HDFs, and thus we next asked whether these interactions might contribute to limit reprogramming TFs ability to remodel the chromatin during the fate conversion process[40–43]. To address this question, we first generated single-cell (sc)

chromatin accessibility profiles from HDFs, 2 days after cardiac reprogramming TFs (MGT) overexpression in siControl- or siAJSZ-transfected HDFs using scATAC-seq. t-distributed stochastic neighbor embedding (t-SNE) clustering of HDFs in MGT + siControl cells (15,859 cells) revealed that cells were distributed as a compact continuum of clusters (Fig. 4a), as in HDFs at ground state (Supplementary Fig. 9a), thus indicating that MGT overexpression alone did not induce major chromatin accessibility profile differences in HDFs, which is consistent with previous observations showing that MGT alone is not sufficient to induce direct reprogramming in human fibroblasts[44,45]. In contrast, t-SNE clustering of MGT + siAJSZ cells (8966 cells) identified a discrete cell population (*cluster 2*) with a chromatin accessibility profile that significantly diverged from the remaining cell populations (black arrow, Fig. 4b). This cluster represents ~13% of total cells, which is notably similar to the percentage of ACTN2 + iCMs (~16%) generated after siAJSZ transfection and MGT overexpression in the CR assay (see Fig. 3c), and represents ~43–52% of MGT overexpressing cells (~25–30% of HDFs overexpress MGT, see Supplementary Fig. 3a, b). Next, to define whether cells from *cluster 2* were undergoing reprogramming, we performed an ontology analysis for genes with differentially accessible (DA) transcriptional start sites (TSSs) in *cluster 2* as compared to *clusters 1* and *3–7*, and observed a 5–10-fold enrichment ($p < 0.0001$) for cardiac terms (i.e., striated muscle contraction and myofibril assembly) (Fig. 4c and Supplementary Data 9), involving a wide array of cardiac-specific genes (i.e., *NKX2-5*, *ACTA1*, and *NPPA*, Supplementary Fig. 9b). Collectively these results indicate that (1) *cluster 2* represents a cell population with a chromatin accessibility profile indicative of cells undergoing fate reprogramming towards the cardiac lineage and (2) AJSZ regulate chromatin accessibility dynamics during this process.

To further delineate the role of AJSZ in the regulation of chromatin accessibility during reprogramming, we mapped all regions of DA chromatin in unchallenged and reprogramming resistant (= all cells from MGT condition) HDFs as compared to cells undergoing fate reprogramming in MGT + siAJSZ condition (= *cluster 2*). This approach identified two domains where the chromatin was specifically open in unchallenged and reprogramming resistant HDFs and closed in cells from *cluster 2* (*domain 1*) or the converse (*domain 2*). Remarkably, these two domains were scattered across the genome (Fig. 4d) and consisted of short DNA regions (<600 bp) (Fig. 4e and Supplementary Fig. 9c), collectively spanning ~7 Mbp for *domain 1* and ~ 4.5 Mbp for *domain 2*. Next, to characterize potential molecular differences between these two domains, we performed a motif analysis and observed a ~4-fold enrichment for AP-1 TFs motifs (i.e., JUNB putative binding site, 1 motif/kb) in *domain 1* as compared to *domain 2* (Fig. 4f, Supplementary Fig. 9d, and Supplementary Data 10). In contrast, in *domain 2*, binding sites for MEF2 TFs were most differentially enriched (~3.5-fold, 0.6 motif/kb) (Fig. 4f) and included canonical binding sequence for cardiac reprogramming TF, MEF2C (Supplementary Fig. 9e, Supplementary Data 10, and ref. [46]). Finally, to determine whether AJSZ might directly interact with *domain 1* and *2*, we quantified the number of AJSZ binding sites at these regions in unchallenged HDFs. Remarkably, this analysis revealed that JUNB was the major interactor for both domains (Fig. 4g, h), thus indicating that JUNB might play a direct role in the maintenance of *domain 1* in an open state

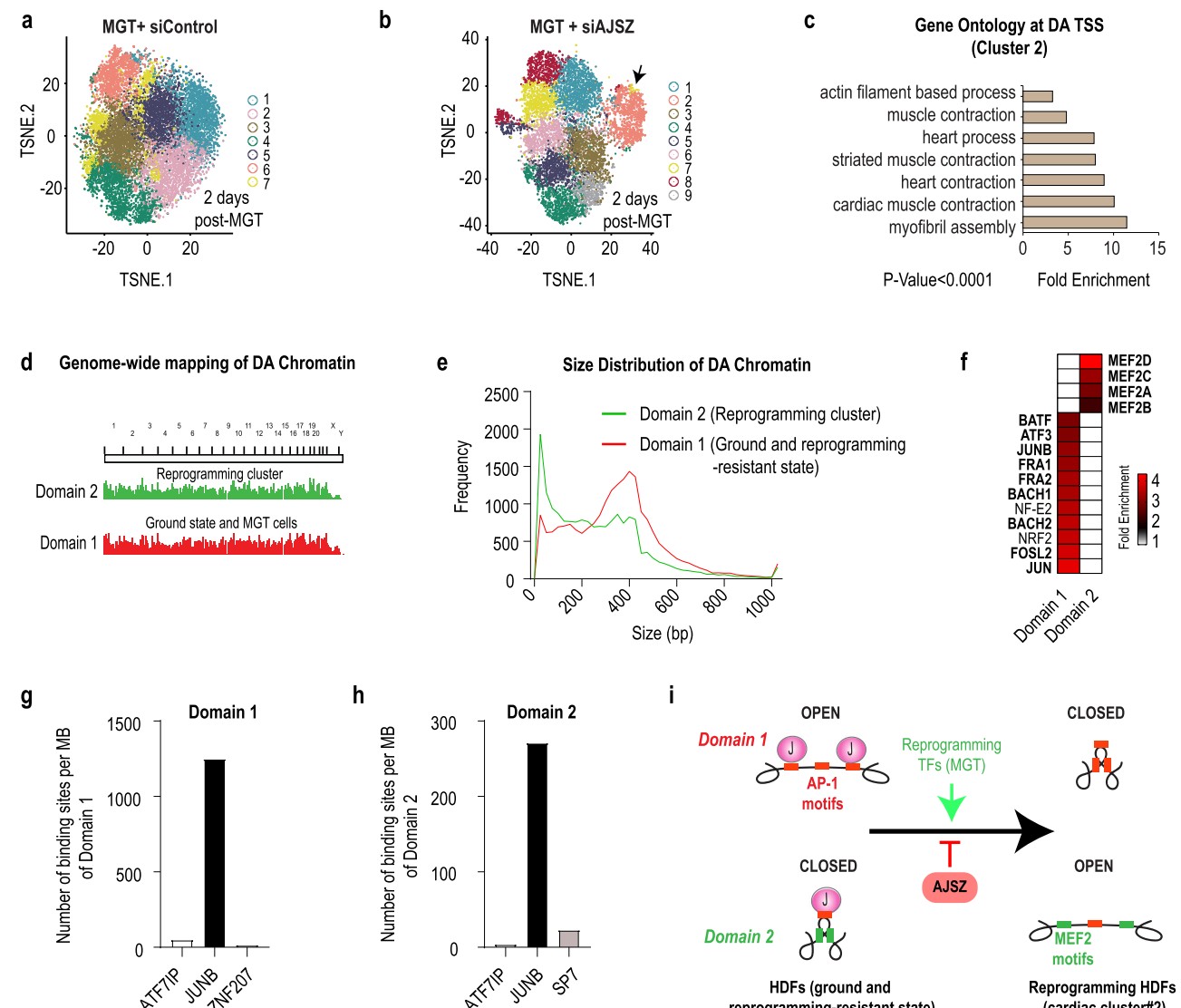

**Fig. 4 | AJSZ regulate chromatin accessibility dynamics during cell fate reprogramming. a, b** t-SNE visualization of cell clusters after scATAC-seq of siControl-transfected HDFs 2 days after MGT overexpression (**a**), or siAJSZ-transfected HDFs 2 days after MGT overexpression (**b**). **c** GO term analysis for genes with differential accessibility transcriptional start sites (TSS) in *cluster 2* vs remaining 8 clusters. **d** Topological mapping of domains 1 and 2 in regard to their chromosomal location.

**e** Size distribution of differentially accessible chromatin regions in domains 1 and 2. **f** Most differentially enriched TF motifs in domain 1 as compared to 2 and conversely. **g, h** AJSZ binding density in domain 1 (**g**) and 2 (**h**) at ground state in HDFs. **i** Model summarizing the role of AJSZ in the regulation of chromatin accessibility during cell fate reprogramming.

and *domain 2* in a closed state at ground state in HDFs. In sum, our data show that AJSZ contributes to regulate chromatin accessibility via direct binding of JUNB to AP-1 motif-enriched chromatin, thereby limiting the number of motifs accessible to the reprogramming TFs and consequently restricting their ability to bind their target DNA and promote cell fate conversion (Fig. 4i).

## AJSZ-mediated control of transcription during cell fate reprogramming

TFs are proximal regulators of transcription[25,47] and consistent with this role, our ChIP-seq data also revealed that AJSZ bind to >9300 promoter-TSS regions (−1 and +0.1 kbp) in HDFs at ground state (see Supplementary Data 11). The magnitude of this interaction led us to postulate that, in addition to their role in the regulation of chromatin accessibility, AJSZ might also oppose cell fate reprogramming via the proximal regulation of transcription. To test this hypothesis, we performed genome-wide RNA-seq of control and AJSZ KD HDFs 2 days after cardiac reprogramming TFs (MGT) overexpression. We identified

736 differentially expressed (DE) genes, of which 501 and 235 were downregulated and upregulated, respectively, by AJSZ KD ($p < 0.05$, Supplementary Data 12 and Fig. 5a). Integration of RNA- and ChIP-seq datasets revealed that ~2/3 of the DE genes (460 of 736, see Supplementary Data 13) were bound at their core promoter regions by at least one of the four fate-stabilizing TFs in HDFs (Fig. 5b). Notably, core promoter binding correlated with gene downregulation for ~75% of DE genes, thus indicating a predominantly activating role for AJSZ in the regulation of transcription during cell fate reprogramming. In this context, GO term analysis of core promoter-bound DE genes revealed a significant enrichment of terms related to cell fate specification, cardiac muscle differentiation, fibroblast proliferation, collagen organization, and TGFβ signaling, thereby supporting a potential role for AJSZ in the control of cell fate-regulating transcriptional programs in fibroblasts (Fig. 5c and Supplementary Data 14). Moreover, assessment of individual AJSZ contributions to core promoter binding revealed that 97% of the core promoters were bound by JUNB in HDFs (Fig. 5d). In this context, analysis of JUNB binding site distribution showed that

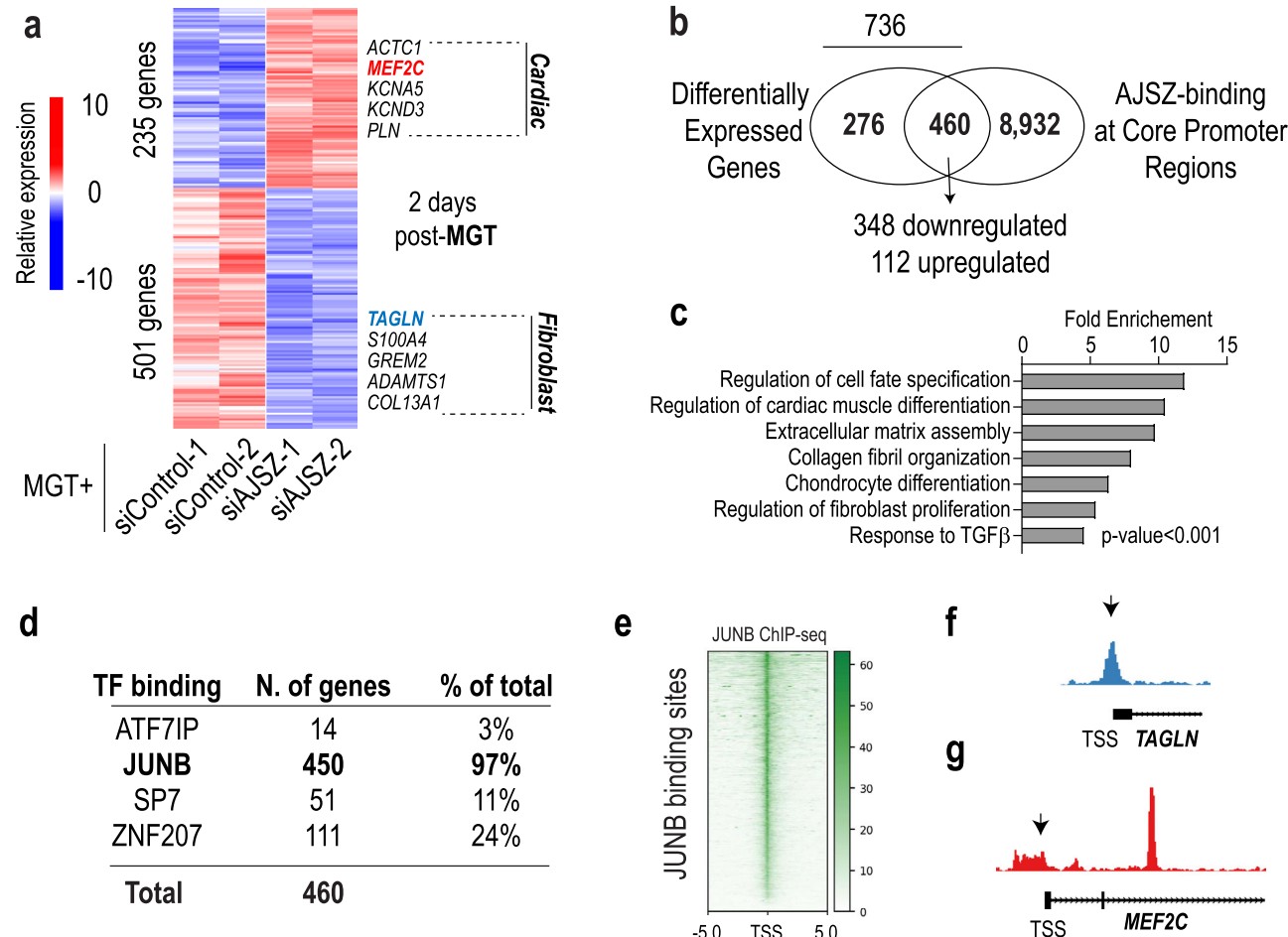

**Fig. 5 | AJSZ proximally regulates gene expression during cell fate repro-gramming. a** Heatmap of differentially expressed (DE) genes in siControl- and siAJSZ-transfected HDFs 2 days after MGT overexpression **b** Venn diagram showing the overlap between DE and core promoter-bound (−1 kb-TSS- +0.1 kb) genes. 460 genes were both DE and bound by AJSZ at core promoter regions, including 348 downregulated and 112 upregulated genes. **c** Bar charts showing top-ranked biological terms enriched for the 460 DE and core promoter-bound genes. **d** Breakdown of the percentage of DE and core promoter-bound genes containing ATF7IP, JUNB, SP7, or ZNF207 binding sites. **e** ChIP-seq tracks for JUNB binding sites. **f, g** Genome browser views showing JUNB binding at TAGLN (**f**) and at MEF2C core promoter regions (**g**) in HDFs.

these interactions were centered at the TSS (Fig. 5e) and could be observed for both downregulated (*TAGLN*) and upregulated (*MEF2C*) genes (Fig. 5f, g). In sum, our results show that AJSZ plays a proximal role in the regulation of fibroblast (TGFβ, collagen organization, pro-liferation) and cell fate-modulating transcriptional programs during reprogramming, at least in part via JUNB binding at core promoter/TSS regions.

**AJSZ-regulated gene network controls cell fate reprogramming**
Given that AJSZ acts both as fate stabilizers and proximal regulators of transcription, we next postulated that they might promote cell fate stability by modulating the expression of downstream reprogramming barriers and agonists (Fig. 6a, e). To explore this hypothesis, we tested the top 25 percentile of core promoter-bound and downregulated genes (MGT + siAJSZ vs MGT + siControl) for the barrier to repro-gramming function, using a siRNA-mediated KD strategy in the iMGT-MEFs assay. Consistent with our hypothesis, this approach identified two hits, siChst2, and siNceh1, that robustly enhanced CR efficiency (Fig. 6b–d and Supplementary Data 15), thus uncovering carbohydrate sulfotransferase 2 (*Chst2*), which mediates 6-*O* sulfation within proteoglycans[48], and neutral cholesterol ester hydrolase 1 (*Nceh1*) which regulates lipid droplet formation[49] and platelet-activating factor synthesis[50], as barriers to CR.

Next, to identify AJSZ-regulated reprogramming agonists (Fig. 6e), we tested the top 25 percentile core promoter-bound and upregulated genes (MGT + siAJSZ vs MGT + siControl) for functional requirement (siRNAs) in the siAJSZ-induced CR assay. This approach identified 61 siRNAs that blunted siAJSZ-induced CR by at least 50% (Fig. 6f and Supplementary Data 16). Of these, only nine siRNAs did not affect cell viability (Supplementary Data 16) and were selected for further analysis (Fig. 6g, h). Remarkably, the nine agonists could be grouped into four ontologies: (1) cell fate modulation (Mef2c)[46], (2) protein folding and degradation (Emc1, Hspb3, Ppic, and Tpp1)[51–53], (3) signaling pathway regulation (Il7r, Olfml3, and Tcta)[54–56], and (4) energy homeostasis (Efhd1)[57].

Finally, we asked whether the AJSZ-mediated regulation of reprogramming agonists and barriers, was lineage and/or cell type-specific. Remarkably, expression analysis revealed that siAJSZ led to a lineage- and cell type-independent upregulation of reprogramming agonists (*MEF2C*, *TPP1*, *PPIC*, *IL7R*, and *EFHD1*) after 3 days of reprogramming (Fig. 6i). Collectively, these results show that a conserved component of AJSZ-mediated reg-ulation of cell fate stability, is the proximal downregulation of genes required for large-scale phenotypic changes (i.e. cell fate modulation, proteome remodeling, energy homeostasis, and inflammation signaling) (Fig. 6j).

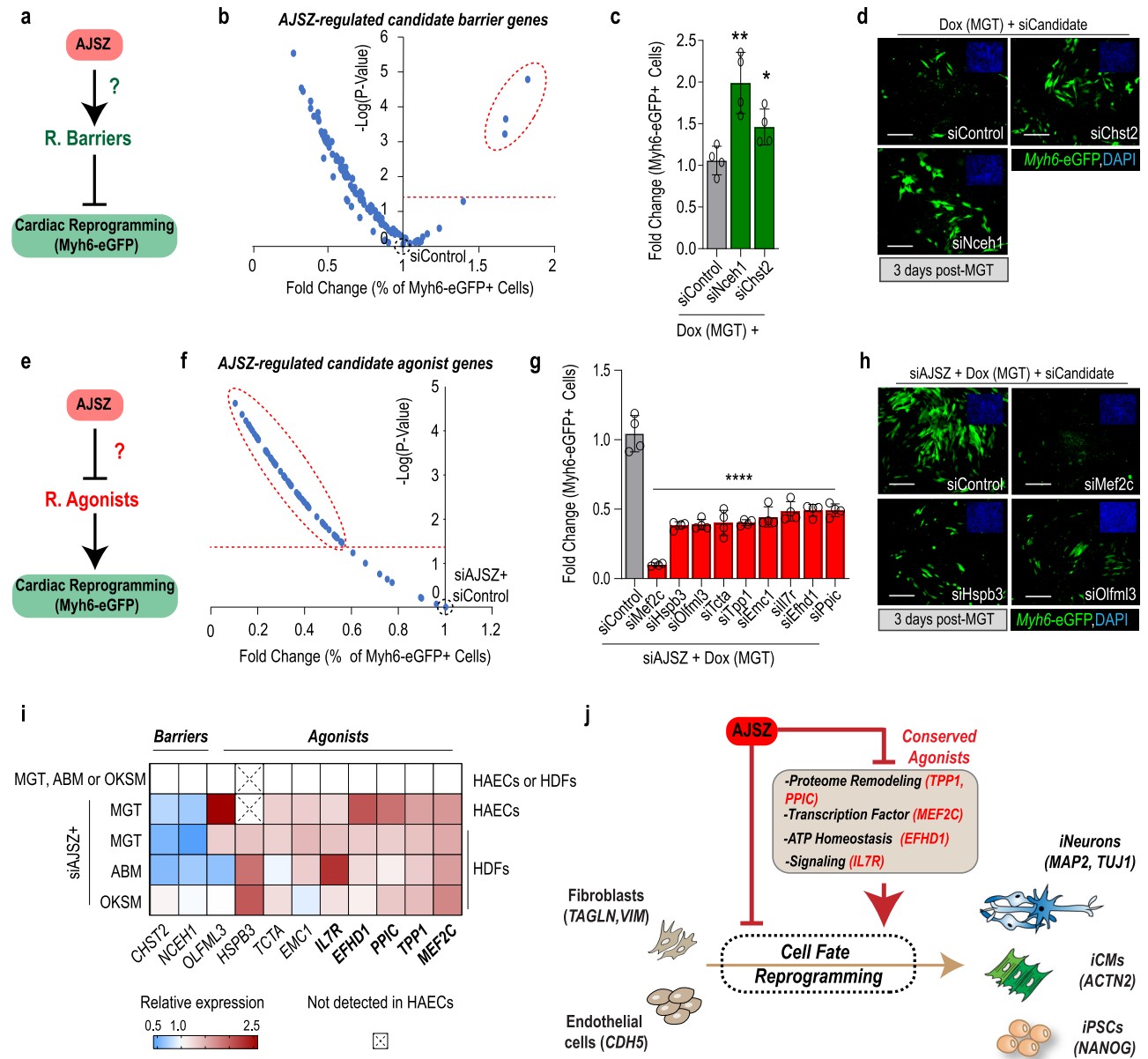

**Fig. 6 | AJSZ control reprogramming barriers and agonists expression.**
**a** Schematic depicting the hypothesis that AJSZ promotes the expression of reprogramming barriers. **b** Volcano plot showing the screening results for siRNAs directed against top 25 percentile downregulated (MGT + siAJSZ vs MGT) and core promoter-bound genes in the iMGT-MEF CR assay. The top three candidate barrier genes are circled. $n = 4$ per condition. **c** Histogram showing validation of siNceh1 and siChst2 effect on CR. $n = 8$ per condition. **d** Representative images for siControl, siChst2, and siNceh1 conditions. Myh6-eGFP+ cells are shown in green and cell nuclei are stained blue (DAPI, top right insets). **e** Schematic depicting the hypothesis that AJSZ negatively regulates reprogramming agonists. **f** Volcano plot showing the screening results for siRNAs directed against the top 25 percentile of upregulated (MGT + siAJSZ vs MGT) and core promoter-bound genes

in siAJSZ-induced iMGT-MEF assay. **g** Histogram showing validation of top nine siRNAs that blunt siAJSZ-induced CR without affecting cell viability. $n = 4$ per condition. **h** Representative images for siAJSZ+ siControl, siMef2c, siHspb3, or siOlfml3 conditions. **i** Heatmap summarizing AJSZ expression dependence of identified barriers and agonists in HAECs and HDFs, 2 days after MGT, ABM, or OKSM overexpression. **j** Model showing that AJSZ regulates cell fate reprogramming by controlling the expression of a conserved set reprogramming agonists. Scale bars: 50 μm. Student's $t$-test. *$p < 0.05$, **$p < 0.01$, and ****$p < 0.0001$. Groups were compared using two-tailed unpaired analysis. Data in the figure are presented as mean values ± standard deviation. **j** Schematic is modified from Cunningham, T. J. et al. Id genes are essential for early heart formation. Genes & development, https://doi.org/10.1101/gad.300400.117 (2017). - CC-BY 4.0.

## shAJSZ enhance MGT-mediated improvement of heart function after myocardial infarction

Retrovirus-mediated overexpression of MGT has been shown to induce CR in vivo and, as a result, improve heart function post-myocardial infarction (MI)[58]. Thus, given the established role of AJSZ as barriers to CR in vitro (see Figs. 1, 2), we next asked whether targeting AJSZ could enhance MGT-mediated CR and further improve heart function post-MI. First, we verified that the loss of AJSZ function is

compatible with normal cardiomyocyte (CM) function. Indeed, transfection of siAJSZ in hiPSC-derived ventricular-like CMs[34,59] did not affect cardiac contractility parameters such as calcium transient kinetics (i.e., calcium transient duration) and/or beating frequency as compared to siControl (Supplementary Fig 10a−d and Supplementary Movies 2, 3). Next, we assessed AJSZ expression in the infarcted region and observed that the four factors are expressed in the injured fibrotic (Col1+) compartment, while Atf7ip and Zfp207 are also expressed in

cardiac (Tnnt2+) compartment (Supplementary Fig. 11a). Finally, we confirmed that co-infection of iMGT-MEFs with shRNAs targeting AJSZ could enhance CR (Supplementary Fig. 11b, c).

To assess whether shAJSZ could enhance MGT's ability to improve heart function post-MI, we delivered PBS or virus mixtures mediating MGT overexpression alone (MGT) or MGT overexpression combined with AJSZ KD (MGT + shAJSZ) by ultrasound-guided injection[60] at the site of injury, 3 days post-MI (Fig. 7a). We confirmed the efficacy of in vivo KD by evaluating Atf7ip and Zfp207 expression levels at the site of injury, 4 days post-injection (Supplementary Fig. 12a–i). 25 days post-injection, we quantified scar size by Masson trichrome staining and observed a ~40% reduction of the scar area in MGT + shAJSZ as compared to MGT (Fig. 7b, c), and a significant decrease in the percentage of Col1+ expressing cells (Fig. 7d, e). In addition to scar size reduction, the cardiac content (% of Tnnt2+ cells) of sections from the MGT + shAJSZ condition was increased by >2-fold as compared to the MGT condition (Fig. 7f, g). Thus, collectively, our data suggest that shAJSZ enhances MGT-mediated CR efficiency in vivo, although a direct demonstration of this process will require genetic labeling of fibroblasts prior to the induction of cell fate reprogramming[58]. In this context, we assessed heart function by quantifying ejection fraction (EF) and fractional shortening (FS) of the left ventricle. Remarkably, EF and FS were both improved by ~100% in MGT + shAJSZ-injected hearts as compared to PBS treatment (EF from 11.4 to 23.3%, FS from 5 to 10.8%) and ~50% as compared to MGT (EF from 15.8 to 23.3%, FS from 7 to 10.8%) at both 2 and 4 weeks after MI (Fig. 7h, i and Supp Fig. 13a–c). Thus, collectively, our results show that shAJSZ enhances MGT's ability to improve heart function post-MI and, thus, suggest that the targeted inhibition of fate stabilizers represents a promising strategy to improve reprogramming-induced heart repair post-injury.

## Discussion

Here, we report on the identification of four TFs (AJSZ) that promote cell fate stability and oppose cell fate reprogramming in both a lineage- and cell type-independent manner. A detailed analysis of their mode of action reveals that AJSZ opposes reprogramming by concomitantly limiting chromatin accessibility dynamics and restricting the expression of genes required for large-scale phenotypic changes. Remarkably, shRNA-mediated targeting of AJSZ post-myocardial infarction was sufficient to reduce scar size and enhance MGT-induced heart function, thus, identifying fate stabilizers as a promising class of targets to enhance adult organ repair post-injury.

### Identification of a generic mechanism regulating cell fate stability in differentiated cells

Given the diversity of cell types co-existing in multicellular organisms and the necessity for these cells to maintain phenotype to fulfil specialized functions, a central point to discuss, is whether the role of fate stabilizers, such as AJSZ, is generic to all cell types or specific to a subset of differentiated cells and/or lineages. Our results generated in multiple reprogramming assays (Figs. 1, 2) show that the role of AJSZ as barriers to reprogramming is conserved across species (mouse and human), cell types (fibroblasts and endothelial cells) and lineages (cardiac, neural, and iPSCs). Moreover and consistent with a generic fate-stabilizing role for AJSZ, these fate stabilizers are expressed in most adult tissues (https://www.proteinatlas.org/)[38], while their expression is reduced in fate-destabilized cells from cancers of distinct lineages (i.e., blood, bone, and prostate)[61–65]. Consistent with these observations, profiling of AJSZ expression in hiPSCs revealed that JUNB and SP7 are not expressed in undifferentiated cells (Supplementary Fig. 14a), while their differentiated progeny robustly expresses all four factors (Supplementary Fig. 14b). Thus, given their conserved role as barriers to cell fate reprogramming, elevated expression in differentiated cells and reduced expression in fate-destabilized cancer and undifferentiated cells, we propose that AJSZ contribute to establish a

generic mechanism promoting cell fate stability in differentiated cells. In this context, we hypothesize that regulators of AJSZ expression and/or activity, might represent yet-to-be-identified factors mediating phenotypic stability in differentiated cells.

### Integrated control of chromatin accessibility and transcription during cell fate reprogramming

TFs recognize specific DNA sequences to control chromatin architecture[42] and transcription[25], and in this study we explored how fate-stabilizing TFs (AJSZ) integrate these two regulatory dimensions to mediate cell fate stability. Our detailed analysis (ChIP-, scATAC-, and RNA-seq and Figs. 3–5) of the AJSZ mode of action, revealed that AJSZ exerts their fate-stabilizing role via direct DNA binding to three distinct chromatin regions. The first region involves ATF7IP, JUNB, and ZNF207 binding to AP-1 motif-enriched open chromatin, where this interaction fulfills two distinct roles: (1) to proximally regulate the expression of lineage-appropriate (i.e., TGFβ, collagen organization, and proliferation in HDFs) and reprogramming-regulating genes and (2) to maintain a subset of ATF7IP, JUNB, and ZNF207-bound chromatin (*domain 1*) in an open state. Although, more work will be needed to understand how ATF7IP, JUNB, and ZNF207 binding to *domain 1* contributes to maintain these regions in an open state and oppose cell fate reprogramming, recent work from ref. [40], have shown that appropriate closing of AP-1 motif-enriched chromatin is required for efficient reprogramming of MEFs into iPSCs. The second main region of interaction involves JUNB binding to reprogramming TF motifs-enriched (i.e., MEF2C, ASCL1, KLF4, and cMYC) closed chromatin (Fig. 3). In this context, our comparative chromatin accessibility analysis (siControl vs siAJSZ) during cardiac reprogramming of HDFs (Fig. 4), shows that under control conditions, pre-existing binding of JUNB to reprogramming TF (MEF2C) motif-enriched regions (*domain 2*), remains in a closed state even when MGT is overexpressed, thus contributing to limit reprogramming TF access to target its DNA and thereby opposing cell fate conversion. Conversely, reduced AJSZ levels enable the de novo opening of *domain 2* and the generation of cells with a cardiac-like chromatin accessibility profile. In this context, it remains to be established how JUNB binding to *domain 2* contributes to maintain these regions in a closed state. Finally, a third type of interaction involves ATF7IP and SP7 binding at regions of closed chromatin enriched for STAT4/5/6 motifs. Interestingly and consistent with a barrier to cell fate reprogramming role (see Fig. 3h, i), previous studies have shown STAT signaling inhibition increases Ascl1-induced transdifferentiation of glial cells into neurons and improves regeneration in adult mouse retina[66]. In this context, we suggest that ATF7IP and SP7 binding to STAT motif-enriched closed chromatin contribute to mediate a STAT4/5/6-driven mechanism opposing cell fate reprogramming. In sum, we propose that fate-stabilizing TFs via motif-specific (i.e., AP-1 and STAT4/5/6) and regionalized binding to both open and closed chromatin, enable the concomitant maintenance of lineage-appropriate chromatin accessibility and transcription required to maintain phenotype in differentiated cells.

### Upstream and downstream regulators of cell fate stability

The identification of cell fate regulators and, most importantly, their assembly into a pathway mediating phenotypic stability is essential for (1) our understanding of how cells normally resist cell fate challenges (i.e., oncogenic transformation and virus infection) and (2) the development of innovative strategies to improve therapeutic reprogramming.

In this study, the systematic evaluation of TFs as barriers to reprogramming led us to identify Atf7ip as the hit concentrating most of the barrier activity (~ 4-fold) (see Fig. 1c), thus indicating that it might play an upstream role in the regulation of cell fate stability. Molecularly, Atf7ip has been shown to regulate protein levels via both transcriptional[67] and post-translational mechanisms[68]; thus, we

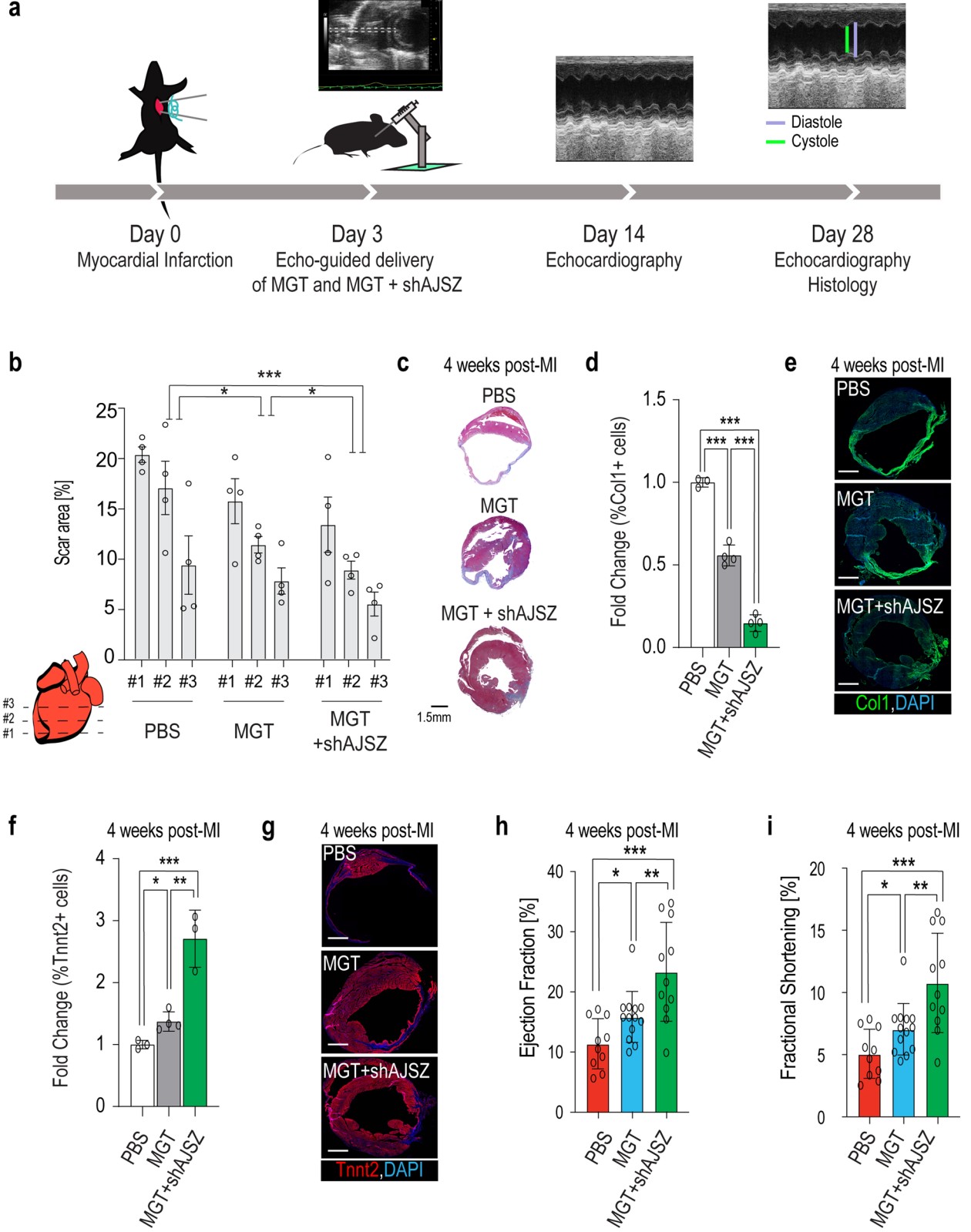

**Fig. 7 | shAJSZ enhances MGT's ability to improve heart function post-MI.**
**a** Schematic depicting the experimental strategy to test the role of AJSZ on heart function post-MI. **b, d, f** Quantification of scar area (**b**), fibrosis (**d**), and cardiac content (**f**) in PBS, MGT, and MGT + shAJSZ conditions 4 weeks after MI. $n = 4$ per condition. **c, e, g** Representative histological heart sections after Masson trichrome staining (scar area) Scale bar: 1.5 mm (**c**), Col1 (fibrosis) (**e**), and Tnnt2 (cardiac content) (**g**) staining in PBS, MGT, or MGT + shAJSZ conditions. DAPI is shown in blue. **h** Ejection fraction (EF) and **i** fractional shortening (FS) of the left ventricle

were serially quantified by echocardiography in mice injected with PBS, MGT, and MGT + shAJSZ 4 weeks after MI. Cardiac function was improved with MGT + shAJSZ as compared to PBS or MGT conditions. For EF and FS quantification: PBS-treated mice $n = 10$, MGT-treated mice $n = 13$, and MGT + shAJSZ treated mice $n = 11$. Groups were compared using two-tailed unpaired. Scale bars: 1.5 mm. *$P < 0.05$, **$P < 0.01$, and ***$P < 0.001$. Data in the figure are presented as mean values ± standard deviation.

propose that it might exert its upstream role by modulating other barrier TFs protein levels. Consistent with this hypothesis, our data (Supplementary Fig. 15a, b) show that KD of Atf7ip causes a 20–50% reduction of Junb, Sp7, and Zfp207 protein levels, while in contrast, KD of Junb does not affect the other barrier TFs. Collectively, our data indicate that Atf7ip plays an upstream role in the regulation of cell fate stability by controlling Junb, Sp7, and Zfp207 protein levels (Supplementary Fig. 15c).

Moreover, the functional evaluation of AJSZ-regulated genes led to the identification of a conserved network of reprogramming agonists (Fig. 6j). A remarkable property for these genes, is their functional requirement for the reprogramming process, even in the context of AJSZ KD, thus implying that they contribute to a non-redundant fate-stabilizing mechanism downstream of the barrier TFs. At the molecular level, these conserved agonists regulate cellular functions mediating protein folding and degradation (PPIC and TPP1), cell fate modulation (MEF2C), energy homeostasis (EFHD1), and inflammation signaling (IL7R), which in turn, limit cells' ability to undergo large-scale phenotypic changes.

### Fate stabilizers, a promising target space for adult organ repair?

The use of cell fate reprogramming to promote adult organ repair is a promise of regenerative medicine[69]; however, to date, low reprogramming efficiency has limited application to the clinic[70]. In this study, we aimed at evaluating whether inhibition of fate-stabilizing TFs, might represent an efficient strategy to enhance direct reprogramming-mediated adult organ repair post-injury. Remarkably, our in vivo data show that shAJSZ significantly enhances MGT's ability to reduce the scar size, increase cardiac content and, most importantly, improve heart function by ~50% post-injury, although, more work is needed to describe the cell types and cellular mechanisms mediating this improvement. In conclusion, our work strongly suggests that the targeted inhibition of fate stabilizers represents a promising strategy to improve direct reprogramming-induced organ repair.

## Methods

### iMGT-Myh6-eGFP-MEFs and screening assays

Immortalized Dox-inducible iMGT-Myh6-eGFP-MEFs were described previously in ref. [32]. The cells were cultured in plates precoated with 0.1% gelatin (Stem Cell Technologies, 7903) and maintained in Fibroblast Culture Medium (FCM) consisting of DMEM (Corning, 10-013-CV), 10% fetal bovine serum (FBS; VWR, 89510-186), and 1% penicillin/streptomycin solution ((10,000 U/mL), Catalog #: 15140122) at 37 °C in a 5% $CO_2$ atmosphere. One day prior to siRNA transfection (day −1), cells were detached by addition of 0.25% trypsin-EDTA (Thermo Fisher, 25200056) for 3 min at 37 °C, and then washed in FCM, centrifuged, and resuspended in Induced-CM Reprogramming Medium (iCRM), consisting of DMEM, 20% Medium 199 (Gibco, 11150-059), 10% FBS, and 1% P/S. Cells were plated in 384-well plates at $10^3$ cells/well and transfected with an siRNA library directed against 1435 mouse TFs (Dharmacon-Horizon Discovery; siGenome-siRNA library, G-015800). The next day (day 0), 1 μg/mL doxycycline hydrochloride (Dox; Sigma, D3072) diluted in iCRM was added to the cells to induce MGT expression. On day 3, the cells were fixed with 4% paraformaldehyde (PFA) and processed for immunostaining, microscopy, and Myh6-eGFP quantification. The top 20 siRNA hits were validated using independent siRNAs (Dharmacon-Horizon Discovery; ON-Target plus pooled siRNAs). All possible combinations of the top eight siRNAs (255 combinations) were assembled by echo-spotting using an Echo 550 liquid handler (Labcyte) and the cells were processed as described above. All experiments were performed in quadruplicate. For follow-up analyses, the cells were collected on 3 after Dox addition for qRT-PCR analysis.

### Human primary dermal fibroblasts (HDFs)

Newborn human primary foreskin fibroblasts were obtained from American Type Culture Collection (ATCC; CRL-2097, CCD-1079Sk) and cultured in plates precoated with 0.1% gelatin in FCM. For cardiac reprogramming, the cells were allowed to reach 80% confluency, harvested using trypsin-EDTA as described above, resuspended in iCRM, added to 384-well plates at $10^3$ cells/well, and transfected with the indicated siRNAs. The next day (day 0), the cells were transduced by the addition of 1 μL/well of mouse *MGT* retrovirus[32] diluted in iCRM. Cells were collected on day 2 for RNA-seq and scATAC-seq experiments, on day 3 days for qRT-PCR experiments, and on day 30 for calcium handling assays. Immunostaining was performed on the days indicated in the legends. During the incubations, 50% of the iCRM medium was exchanged every other day up to day 8, and was then replaced with RPMI 1640 (Life Technologies, 11875093), 1% B27 supplement (Life Technologies, 17504044), and 1% P/S. Neuronal reprogramming was induced as previously described in refs. [7,38]. In brief, HDFs were cultured and harvested as described above, resuspended in FCM, added to 384-well plates at $3 \times 10^3$ cells/well, and transfected with siRNAs. The next day (day −1), 0.25 μL of F-ABM lentiviral mix (1:1:1:1 of Addgene plasmids #27150, Tet-O-FUW-Ascl1; #27151, Tet-O-FUW-Brn2; #27152, Tet-O-FUW-Myt1l; #20342, FUW-M2rtTA) was added to each well, and the cells were incubated for an additional 24 h. The following day (day 0), ABM expression was induced by the addition of 2 μg/mL Dox diluted in FCM. On day 2, media was replaced by Minimal Neuronal Medium consisting of DMEM/F-12 (Gibco, 11220-032), 1% B27, 1% N2 Supplement (Gibco, 17502-048), and 1% human recombinant insulin and zinc solution (Gibco, 12585-014). On day 3, cells were harvested and processed for immunostaining or qRT-PCR analysis. Induced pluripotent stem cell reprogramming was induced as previously described in ref. [41]. In brief, HDFs were cultured and harvested as described above, resuspended in FCM, added to 384-well plates at $1 \times 10^3$ cells/well, and transfected with siRNAs. Cells settled to the bottom of the well for 5 min at room temperature and then 0.25 μL of F-OKMS lentiviral mix (1:1 of Addgene plasmids #51543, FUW-tetO-hOKMS; #20342, FUW-M2rtTA) was added to each well, and the cells were incubated for 24 h. The following day (day 0), OKMS expression was induced by the addition of 2 μg/mL Dox diluted in FCM. On days 2, 4, and 6, media was replaced with Stem Cell Technologies mTeSR Plus Kit (Stem Cell Technologies #100-0276). On day 7, cells were harvested and processed for immunostaining or qRT-PCR analysis of pluripotency genes.

### Human adult aortic endothelial primary cells (HAECs)

Human adult aortic endothelial cells (HAECs) were obtained from ATCC (PCS-100-011) and cultured in plates precoated with 0.1% gelatin in Vascular Cell Basal Medium (VCBM; ATCC, PCS-100-030) containing endothelial cell growth kit-BBE (ATCC, PCS-100-040) and 0.1% P/S. When the cells reached 80% confluency, they were harvested with trypsin-EDTA solution for primary cells (ATCC, PCS-999-003) for 3 min at 37 °C, washed with trypsin-neutralizing solution (ATCC, PCS-999-004), resuspended in VCBM/kit-BBE medium, added to 384-well plates at $3 \times 10^3$ cells/well, and transfected with siRNAs. One day later (day 0), cells were transduced by the addition of 1 μL/well of mouse MGT retrovirus. The cells were collected on day 3 for qRT-PCR analysis and on day 20 for immunostaining. During the incubation, 50% of VCBM/kit-BBE medium was exchanged every other day starting on day 4.

### siRNA transfection

Mouse and human siRNAs were purchased from Dharmacon or Ambion and added at a final concentration of 25 nM. A negative control siRNA (referred to as siCTR or siControl) was obtained from Dharmacon. Transfection was performed using Opti-MEM (Gibco, 31985070) and Lipofectamine RNAiMAX (Gibco, 13778150) according

to the manufacturer's instructions. siRNA transfection was performed on day −1, unless otherwise noted.

## Immunostaining and fluorescence microscopy

Cells were fixed with 4% PFA in phosphate-buffered saline (PBS) for 30 min and blocked with blocking buffer consisting of 10% horse serum (Life Technologies, 26050088), 0.1% gelatin, and 0.5% Triton X-100 (Fisher Scientific, MP04807426) in PBS for 30 min. Cells were incubated with primary antibodies overnight at 4 °C and followed by secondary antibodies with 4′,6-diamidino-2-phenylindole (DAPI; Sigma, D9542) for 1 h in the dark at room temperature. Cells were washed with PBS between each step. Cells were imaged with an ImageXpress confocal microscope (Molecular Devices) and fluorescence was quantified with Molecular Device software. Experiments were performed in quadruplicate.

Mouse heart sections were fixed with 4% PFA (Alfar Aesar, 43368) for 10 min. Sections were permeabilized and blocked in 20% goat serum (Life Technologies, 16210-072) and 0.3% Triton X-100 (Promega, H5142) in PBS for 1 h. Incubation with the primary antibody was performed overnight at 4 °C. Sections were washed with PBS. Then, the secondary antibody was left for 1 h at room temperature. Nuclei were counterstained with DAPI for 10 min. Slides were mounted with Fluoromount-G (Southernbiotech, 0100-01).

The primary antibodies used were: rabbit anti-ATF7IP (Sigma, HPA023505, 1:200); rabbit anti-ATF7IP (Invitrogen, PA5-54811, 1:200); rabbit anti-JUNB (Abcam, Ab128878, 1:200); rabbit anti-SP7/ OSTERIX (Abcam, Ab22552, 1:500); mouse anti-ZNF207 (Sigma, SAB1412396, 1:500); rabbit anti-TAGLN (Abcam, Ab14106, 1:800); guinea pig polyclonal anti-Vimentin (Progen, GP53, 1:100); mouse anti-VIMENTIN (Santa Cruz Biotechnology, sc-373717, 1:800); goat polyclonal anti-TAGLN (GeneTex, GTX89789, 1:800); rabbit polyclonal anti-TNNT2 (Sigma, HPA017888, 1:100); mouse anti-ACTN2 (Sigma, A7811, 1:800); goat anti-PECAM1 (H3) (Santa Cruz Biotechnology, Sc1506, 1:200); rabbit anti-MAP2 (Abcam, ab32454, 1:200); and mouse anti-TUJ1 (R&D Systems, MAB1195, 1:200); rabbit anti-NANOG (Abcam, ab109250, 1:200); rabbit anti-Collagen I (Abcam, Ab21286, 1:50). Secondary antibodies were: Alexa Fluor 488 goat anti-rabbit IgG (H + L) (Invitrogen, A11008, 1:1000); Alexa Fluor 488 donkey anti-mouse IgG (H + L) (Invitrogen, A21202 1:1000); Alexa Fluor 488 goat anti-Guinea Pig IgG (H + L) (Invitrogen, A 11073 1:100); Alexa Fluor 568 goat anti-rabbit IgG (H + L) (Invitrogen, A11011, 1:200); Alexa Fluor 568 goat anti-mouse IgG (H + L) (Invitrogen, A10037, 1:1000); Alexa Fluor 568 donkey anti-goat IgG (H + L) (Invitrogen, A11057, 1:1000); Alexa Fluor 680 donkey anti-mouse IgG (H + L) (Invitrogen, A10038, 1:1000); and Alexa Fluor 680 donkey anti-rabbit IgG (H + L) (Invitrogen, A10043, 1:1000).

## RNA extraction and qRT-PCR

Total RNA was extracted using Zymo Research Quick-RNA MircoPrep Kit (Zymo Research, R1051) or TRIzol reagent (Invitrogen, 15596026) and chloroform (Fisher Chemical, C298-500) following the manufacturers' recommendations. RNA in the aqueous phase was precipitated with isopropanol, centrifuged, washed with 70% ethanol, and eluted in DNase- and RNase-free water. RNA concentration was measured by Nanodrop (Thermo Scientific). Aliquots of 1 µg of RNA were reverse transcribed using a QuantiTect Reverse Transcription kit (Qiagen, 205314), and qPCR was performed with iTaq SYBR Green (Life Technologies) using a 7900HT Fast Real-Time PCR system (Applied Biosystems). Gene expression was normalized to that of glyceraldehyde 3-phosphate dehydrogenase (GAPDH) for human samples or β-actin (Actb) for mouse samples using the $2^{-\Delta\Delta Ct}$ method. Human and mouse primer sequences for qRT-PCR were obtained from Harvard Primer Bank. Primers were: ATF7IP (#38261961c1), Atf7ip (#34328232a1), JUNB (#44921611c1), Junb (#6680512a1), SP7 (#22902135c2), Sp7 (#18485518a1), ZNF207 (#148612834c1), Zfp207

(#7212794a1), ACTC1 (#113722123c1), MYL7 (#50593014c1), NPPA (#23510319a1), NPPB (#83700236c1), RYR2 (#112799846c1), SCN5A (#237512981c1), TNNI3 (#151101269c1), TNNT2 (#48255880c1), GAD67 (#58331245c2), PVALB (#55925656c2), SYN1 (#91984783c1), vGLUT2 (#215820654c2), MYH6 (#289803014c3), HSPB3 (#306966173c2), OLFML3 (#50593011c1), TCTA (#148922970c1), TPP1 (#118582287c1), EMC1 (#22095330c3), IL7R (#28610150c2), EFHD1 (#237649043b1), PPIC (#45439319c2), NCEH1 (#226423949c2), CHST2 (#344925865c1), Actb (#6671509a1), and GAPDH (#378404907c1), ASCL1 (#190343011c1), BRN2 (#380254475c1), MAP2 (#87578393c1), MYT1L (#60498972c3), TUBB3 (#308235961c1), cMYC (#239582723c3), SOX2 (#325651854c3), KLF4 (#194248076c2), POU5F1(OCT4) (#4505967a1), NANOG (#153945815c1), DPPA2 (#239835766c1), DPPA4 (#144953902c1), and REX1 (ZFP42) (#89179322c2).

## Immunoprecipitation

Nuclear extracts from HDFs were prepared using NE-PER Nuclear and Cytoplasmic Extraction Kit (Life Technology, PI78833). Protein lysates were pre-cleared with desalting columns (Life Technology, PI89890) and buffer exchange with NP40 lysis buffer consisting of 50 mM Tris-HCl, 150 mM NaCl, 1 mM EDTA, and 0.5% NP40) prior immunoprecipitation. About 500 µg of protein lysate was immunoprecipitated using 70 µl protein G magnetic beads (Lifer Technology, 10004D) and NP40 lysis buffer with 1x protease and phosphatase inhibitor cocktails (Lifer Technology, PI78441) per sample, followed by antibody binding to 10 ug ZNF207 (Lifer Technology, PA5-30641), or IgG isotype control (Cell signaling technology, 39003) and incubated overnight at 4 °C on the rotator. On the next day, beads were washed three times with wash buffer consisting of 50 mM Tris-HCl, 150 mM NaCl, and processed for denature elution.

## Western blot

Immunoprecipitated magnetic bead-antigen complexes were eluted with LDS sample buffer (Life Technology, NP007), sample reducing agent (Life Technology, NP0009), and heated at 70 °C for 10 min. Eluted protein samples were run on 7% Tris-Acetate Protein Gels (Life Technology, EA0358BOX) and transferred to nitrocellulose membranes (Life Technology, IB23002). Membranes with transferred proteins were blocked with blocking buffer (Intercept, LI-COR, NC1660556) for 1 h and incubated with a primary antibody of 1:600 ATF7IP (Life Technology, PA5-5481), 1:1000 JUNB (Cell Signaling Technology, C37F9) or 1:1000 ZNF207 (Lifer Technology, PA5-30641) overnight at 4 °C on the rotator. Membranes were washed four times and incubated with fluorescently conjugated goat anti-rabbit secondary antibody (LI-COR 92632211) for 1 h. Membranes were washed four times and proceeded with the Odyssey CXL image system for protein fluorescent detection.

## Retrovirus and lentivirus preparation

Large-scale retrovirus production was performed at the SBP Viral Vector Core Facility SBP as previously described[34]. Briefly, for retrovirus preparation, pMX-MGT[71], Retro-Gag-Pol, and pMD2.G plasmids were co-transfected into HEK-293T cells at a ratio of 3:2:1. For lentivirus preparation, lentivector DNA plasmids (Addgene plasmids #27150, Tet-O-FUW-Ascl1; #27151, Tet-O-FUW-Brn2; #27152, Tet-O-FUW-Myt1l; #20342, #51543, FUW-tetO-hOKMS, FUW-M2rtTA, pLKO.1 shAtf7ip #TRCN0000374251, pLKO.1 shJunb #TRCN0000232241, pLKO.1 shSp7 #TRCN000082147, pLKO.1 shZfp207 #TRCN0000225905 were individually co-transfected with pCMVDR8.74 and pMD2.G into HEK-293T cells using the calcium phosphate method. UltraCulture serum-free medium (Lonza) supplemented with 1 mM ʟ-glutamine (Life Technologies) was used to re-feed transfected cells, and the supernatant was collected every 24 h from day 2 to day 4 after transfection. Viral supernatants were pooled, passed through a 0.22-µm-pore filter, concentrated, and purified by 20% sucrose gradient

ultracentrifugation at 21,000 rpm for 2 h at 4 °C. The pellet containing concentrated viral particles was resuspended in PBS, aliquoted, and kept at −80 °C.

## Calcium handling assay

The calcium assay was performed on day 30 HDFs after MGT over-expression or day 28 hiPSC-CMs post-transfection with siControl or siAJSZ. The assay was performed using the labeling protocol as previously described in ref. [72]. Briefly, 50% of the cell culture supernatant was replaced with a 2X calcium dye solution consisting of Fluo-4 NW dye (Invitrogen, F36206), 1.25 mM probenecid F-127 (Invitrogen), and 100 μg/mL Hoescht 33258 (Invitrogen, H3569, in water) diluted in warm Tyrode's solution (Sigma), and the cells were incubated at 37 °C for 45 min. The cells were then washed four times with fresh pre-warmed Tyrode's solution and automatically imaged with an ImageXpress Micro XLS microscope (Molecular Devices) at an acquisition frequency of 100 Hz for a duration of 5 s with excitation 485/20 nm and emission 525/30 nm filters. A single image of Hoescht fluorescence was acquired before the time series. Fluorescence quantification over time and single-cell trace analysis were automatically performed using custom software packages developed by Molecular Devices and the Colas laboratory.

## RNA-seq and data analysis

HDFs were added to 384-well plates at $10^3$ cells/well and transfected with siRNAs (siATF7IP, siJUNB, siSP7, siZNF207 individually or in combination) in iCRM. The next day (day 0), the cells were transduced with 1 μL/well mouse MGT retrovirus diluted in iCRM. On day 2, cells were collected and RNA was extracted using TRIzol. Cells were pooled from 16 wells per sample, and two biological replicates per condition were analyzed. Library preparation was performed by Novogene using their in-house preparation protocol. Briefly, mRNA was enriched using oligo (dT) beads and fragmented randomly using a fragmentation buffer. cDNA was generated from an mRNA template using a random hexamer primer followed by second-strand synthesis. Terminal repair, A ligation, and sequencing adapter ligation were then performed. The final libraries were generated through size selection and PCR enrichment and sequenced as 2x150bp on a HiSeq2500 Sequencer (Illumina). Samples were sequenced to an approximate depth of 35–40 million reads per sample. Raw sequencing reads were trimmed using Trimmomatic (0.36) with a minimum quality threshold of 35 and a minimum length of 36[73]. Processed reads were mapped to the hg38 reference genome using HISAT2 (2.0.4)[74]. Counts were then assembled using Subread featureCounts (1.5.2)[75]. Differential gene expression was analyzed using the DESeq2 package (1.20)[76] in R. Genes were defined as differentially expressed if the adjusted $p$ value was <0.05 after correction for multiple testing using the Benjamini–Hochberg method. Gene Ontology (GO) analysis was performed using PANTHER version 12.0 classification[77,78].

## Single-cell ATAC-seq (scATAC-seq)

scATAC-seq experiments were performed with control (untreated) HDFs, MGT + siControl HDFs, and MGT + siAJSZ HDFs. HDFs were added to 384-well plates at $2.5 × 10^3$ cells/well in iCRM and transfected with siRNAs. The next day (day 0), cells were transduced to mouse MGT retrovirus and collected 2 days later using trypsin-EDTA. Cells from 40 wells were pooled to obtain $≥2 × 10^5$ cells per sample, with two biological replicates per condition. Cells were washed with FCM, centrifuged in conical tubes, and the pellets were frozen in Freezing Medium (DMEM, 10% DMSO, and 20% FBS), transferred to cryotubes, and placed in Mr. Frosty containers (Thermo Fisher) at −80 °C.

Samples were processed for scATAC-seq at UCSD CMME. Samples were processed for scATAC-seq at UCSD CMME using 10x Genomics and sequenced on a NovaSeq 6000 at a depth of 20–25k read pairs per nucleus. Cell Ranger-ATAC (1.1.0) pipeline was used to filter and align reads, count barcodes, identify transposase cut sites, detect chromatin peaks, prepare t-SNE dimensionality reduction plots, and compare differential accessibility between clusters. The Cell Ranger-ATAC pipeline uses the following tools: cutadapt, BWA-MEM, SAMtools tabix, and bedtools. Further differential accessibility analysis was performed using DiffBind (2.12.1) and custom R scripts and visualized with ggplot2. Tracks were visualized using Integrative Genome Viewer 2.8.12. All scripts for this analysis are available on GitHub [https://github.com/smurph50].

## Chromatin immunoprecipitation-seq (ChIP-seq)

ChIP-seq experiments were performed with $100 × 10^6$ HDFs per replicate using a SimpleChip Plus sonication chromatin IP kit (Cell Signaling Technology, 56383) according to the manufacturer's directions. In brief, HDFs were grown to 80% confluency ($10^7$/sample) and then crosslinked with 1% formaldehyde (Sigma, F8775) in PBS at room temperature for 20 min with occasional stirring. The crosslinking reaction was quenched by the addition of 0.125 M glycine for 10 min, and chromatin was fragmented for 25 min using a Bioruptor Pico sonicator (Diagenode) to an average DNA fragment length of 200–500 bp. DNA was quantified with Qubit (Invitrogen, Q32854). Samples equivalent to 100 μg of DNA were incubated overnight at 4 °C with 4 μg of rabbit polyclonal anti-ATF7IP (Invitrogen, PA5-54811), rabbit monoclonal anti-JUNB (C37F9) (Cell Signaling Technology, 3753 S), rabbit anti-SP7/OSTERIX (Abcam, Ab22552), or rabbit polyclonal anti-ZNF207 (Bethyl laboratories, A305-814AM). Rabbit IgG (Cell Signaling Technology, 2729) was used as a negative control. Immunocomplexes were captured by rotation with protein G-coupled magnetic beads (Cell Signaling Technologies, 9006) for 2 h at 4 °C, and immunoprecipitated genomic DNA was collected by incubation with 50 μL elution buffer. Library preparation and sequencing of immunoprecipitated and input DNA was performed by the UCSD IGM core facility. Raw reads were mapped to GRCh38 using Bowtie2 (2.3.5). Since each sample was run across two lanes, SAM files were merged using Picard (2.20.5). MACS2 (2.1.1) was used to call narrow peaks relative to input with a $q$ value cutoff of 0.01. Peaks were annotated with Homer (4.10.4) and motifs were analyzed using MEME-ChIP (5.1.1). BigWig files were generated using Deeptools (2.2) bamCoverage. Tracks were visualized with Fluff (biofluff 3.0.3). Gene ontology biological process terms were found with PANTHER GO and overlap analyses were performed using custom R scripts with venneuler and ggplot2 packages. Bedtools (v2.29.2) was used for genomic comparisons and combining ChIP-seq and scATAC-seq data. HOMER was used to find motifs with a scrambled background.

## Mouse MI model

Experiments were performed in 10- to 12-week-old randomly allocated male and female mice (Jackson lab, strain 000664). To generate the mouse MI model, mice were intubated and anesthetized with isoflurane gas. The chest cavity was exposed by cutting the intercostal muscle and then the left coronary artery was ligated with a 5-0 silk suture.

## Echo-guided retro/lentiviruses injection

Three days after MI, 6 μL of virus-containing solution (PBS) was injected into the boundary between the infarct and border zone at one site with a 32-gauge needle using echo-guided visualization as described in ref. [60]. The mouse surgeon was blinded to the study and no mortality after MI was noted.

## Echocardiography

Cardiac function was analyzed with transthoracic echocardiography (Visual Sonics, Vevo 2100) at 2 weeks, and 4 weeks after MI ($n = 10$–13 per group). Mice were anesthetized with low-dose isoflurane for echocardiographic examination. Two-dimensional targeted M-mode

traces were obtained at the papillary muscle level. Left ventricular internal diameter during diastole (LVDd) and left ventricular internal diameter during systole (LVDs) were measured in at least three consecutive cardiac cycles. EF and FS were calculated with the Teichholtz formula. The average baseline prior to injury for EF and FS was 78.9 and 45.9%, respectively.

## Statistical analysis

All statistical analyses were performed using Prism version 8.0 (GraphPad Software, San Diego, CA, USA). Data are presented as the mean ± standard deviation unless noted. Statistical significance was analyzed by unpaired Student's *t*-test or one-way ANOVA. *P* values of <0.05 were considered significant.

## Reporting summary

Further information on research design is available in the Nature Portfolio Reporting Summary linked to this article.

## Data availability

The RNA-seq, ChIP-seq, and scATAC-seq data generated in this study have been deposited in the GEO database under accession codes GSE183121, GSE183122, GSE183123, GSE183124. The non-sequencing data generated in this study are provided in the Supplementary Information/Source Data file. Source data are provided with this paper.

## Code availability

All scripts for this analysis are available on GitHub [https://github.com/smurph50 https://doi.org/10.5281/zenodo.7629804 and https://doi.org/10.5281/zenodo.7629826].

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

## Acknowledgements

This work was supported by grants DISC2-10110 (California Institute for Regenerative Medicine), R01 HL153645, R01 HL148827, R01 HL149992, R01 AG071464 (National Institutes of Health), and SBP institutional support to A.R.C. Sanford Burnham Prebys Shared Resources are

supported by an NCI Cancer Center Support Grant (P30 CA030199). We thank Kirsten Jepsen (UCSD IGM Genomic Core Facility) for assistance with the ChIP-seq experiments; Allen Wang and Sebastian Preissl (UCSD CMME) for assistance with the scATAC-seq experiments; Nadan Wang and Michelle Leppo (JHU SOM) for assistance with mouse surgeries and echo-guided injections; Luca Caputo, Haley Vaseghi, and Li Wang for kindly sharing reagents and instruments; and Sean Spiering, Eleanor Kim, and Josiah Punay for excellent technical support.

## Author contributions

M.A.M., S.M., and M.Ly designed, performed experiments, analyzed data, and wrote the paper; M.S.Y., A.K., Y.-L.C., S.K., M.Lo, C.L., P.Am, H.T., and C.-T.H. performed experiments and analyzed data; P.L.P., C.K., P.D.A., L.Q., A.S., P.An, and A.R.C. designed experiments, analyzed data, and wrote the paper.

## Competing interests

The authors declare no competing interests.
