## [Peer Review File · Nature Communications]

Conserved Transcription Factors Promote Cell Fate Stability and Restrict Reprogramming Potential in Differentiated CellsREVIEWER COMMENTS

Reviewer #1 (Remarks to the Author):

Cell fate reprogramming is expected to be a new regenerative medicine, but the low reprogramming efficiency is an issue that needs to be solved. For successful cell fate conversion, the reprogramming factor must bind to the target DNA efficiently through epigenetic regulation or other means to regulate gene expression in the target cell type. However, there has been no systematic evaluation of factors as fate stabilizers in initiating cells. In this paper, to discover novel fate stabilizing regulators, the authors performed a genome-wide TF siRNA screen by applying the cardiac reprogramming assay in mouse embryonic fibroblasts and found that four transcription factors (ATF7IP, JUNB, SP7, and ZNF207 [AJSZ]) robustly inhibited cardiac reprogramming. Furthermore, the authors showed that these four factors are also common factors controlling fate stabilization when reprogramming primary human fibroblasts and adult endothelial cells into cardiomyocytes, neurons, and iPSCs. Mechanistically, analysis using RNA-seq, ChIP-seq, and ATAC-seq revealed that AJSZ binds to both open and closed chromatin genome-wide and locally, limiting access to target DNA for reprogramming factors and chromatin remodeling, as well as by downregulating a series of conserved genes involved in the regulation of cell fate specification, proteome remodeling, ATP homeostasis, and inflammatory signaling, AJSZ limits the ability of cells to undergo large-scale phenotypic changes. Finally, the authors showed that even in vivo, the combination of KD of AJSZ and cardiac reprogramming can promote improvement in cardiac function and a decrease in fibrosis.

The concept is of interest as developing a new therapeutic strategy for reprogramming. However, I don't think the data is conclusive enough to draw a clear conclusion that the knockdown of the four factors robustly promotes reprogramming. They assessed reprogramming efficiency only by the percentage of specific markers but did not evaluate the function of the reprogrammed cells. Furthermore, quantitative evaluation of reprogramming efficiency should be done more precisely. It is also necessary to verify whether cells converted from AJSZ KD cells are equivalent to cells converted by conventional methods. The mechanism of the improvement of cardiac reprogramming in vivo has not been clarified, and more validation is needed.

My comments are as follows:

Major considerations:

1. First, the role of AJSZ needs to be clarified. The authors state that "AJSZ is expressed in most adult tissues but is down-regulated in lipolytic cells from different lines of cancer," but AJSZ was identified in a screen of immature mouse fetal fibroblasts (MEFs). It needs to be clarified whether these 4Fs are expressed in MEFs.
2. Since the cell cycle and reprogramming efficiency are strongly related, and the effect of siAtf7ip, which has been pointed out to be related to the cell cycle, is the largest among AJSZ (Fig 1C), it is necessary to clarify the effect of AJSZ KD on the cell cycle.
3. Although AJSZ is expressed in mature tissues, AJSZ KD promotes both reprogramming to undifferentiated and differentiated cells. Does AJSZ, which was KD before reprogramming, regain its expression in mature tissues after reprogramming, while it remains lost in undifferentiated cells?
4. Although AJSZ KD shows an increase in Myh6-GFP positive cells, accurate quantitative evaluation by IHC is difficult; more accurate quantitative evaluation such as FACS would support the authors' results. The same can be said for other related reprogramming efficiency assessments.
5. Is AJSZ KD relevant for reprogramming to more functionally mature cells? I mean is AJSZ KD relevant for full reprogramming. For example, does KD alter the efficiency of beating in

cardiomyocytes, the induction of cardiomyocytes with clear sarcomeric structures, or the pluripotency in iPS cells?

6. Figure 2A shows that AJSZ is expressed in HDFs, but fibroblasts are a heterogeneous population. It is necessary to quantify the fibroblasts expressing each factor and to quantitatively evaluate the fibroblasts expressing AJSZ. The same can be said for related assessments in HAECs (Supplementary Fig. 4).

7. Supplementary Fig. 2A confirms the effect of siAJSZ, but HDFs should be added as a control; it should be shown whether MGT overexpression is independent of AJSZ expression. The same can be said for related assessments in HAECs (Supplementary Fig. 4).

8. Supplementary Fig. 2B shows that siAJSZ upregulates heart-related gene expression, but is there any change in fibroblast-related genes? It would be better if HDFs were necessary for control. The same can be said for other related assessments (Supplementary Fig. 3).

9. Regarding Figures 2C and 2I, it is difficult to compare the expression of fibroblast markers/endothelial markers and CM markers only with the merged pictures. It is easier to understand if each picture is shown. Moreover, both of them seem to co-express CM markers and fibroblast or endothelial cell markers. Can we confirm that the induced cells are cardiomyocytes?

10. Supplementary Figure 5 appears to contradict the authors' hypothesis that AJSZ is a general cell fate stabilizer. First, the comparison between Figure 5B and Figure 5C is confusing and should be integrated. Also, the reviewers believe that FACS is more accurate than IHC for quantitative evaluation. More critically, if AJSZ acts as a general cell fate stabilizer, it is hard to understand why the reprogramming efficiency decreases with time after AJSZ KD. This discrepancy needs to be verified.

11. The results in Figure 3 show that JUNB and ATF7IP account for most of the binding events. In particular, JUNB not only recruits other factors and associates with open chromatin in the AP-1 motif enriched chromatin region, but is also involved in maintaining the closed state of lineage-specific factors, and seems to be the most important factor. However, Fig. 1C shows that siATF7IP promotes reprogramming efficiency the most. Please explain the reason for this.

12. In fig4B, on t-sne, Cluster2 is a new cluster emerged by AJSZ KD, but does this cluster still exist as a discrete cell population in the integrated dataset of MGT+siCtrl and MGT+siAJSZ?

13. Supplementary Fig. 6B shows Nkx2.5 as a representative of differentially accessible (DA) transcriptional start sites. However, it seems that cardiac reprogramming converts fibroblasts directly into cardiomyocytes without going through cardiac progenitor cells expressing Nkx2.5. Why the change in cardiac stem cell markers, and is it possible that AJSZ KD reprogrammed cardiac stem cells?

14. Other groups have already shown that forced expression of reprogramming factors can change the state of chromatin. In Fig. 4D, Domain 1 is compared with Domain 2 as unchallenged HDFs and HDF, HDFs. Wouldn't it be more appropriate to compare the three groups, HDF, siControl+MGT, siAJSZ+MGT, to evaluate the target of AJSZ?

15. In figure 7, EF after myocardial infarction seems to be too low. It seems to be around 20% in many studies. Was there any difference in survival rate among the three groups? Also, in figure 7E and F, MI + PBS should be shown to evaluate whether the myocardial infarction was properly prepared.

16. Supp Fig.7A alone, it is difficult to show whether AJSZ is indeed expressed in non-cardiomyocytes (especially cardiac fibroblasts). Even if AJSZ is expressed in cardiac fibroblasts, it is necessary to

mention the percentage of fibroblasts expressing AJSZ and the localization of the cells (remote, border, infarct area).

17. Supp Fig.7B,C also need to be quantitatively evaluated by FACS. The shControl in Fig. 7C appears to have almost no GFP+ cells.

18. Supp Fig.8 also needs to show the exact KD efficiency of AJSZ with WB and qRT if possible. In addition, MI+PBS needs to be added as a control in 8A.

19. It is necessary to show whether the efficiency of in vivo reprogramming from fibroblasts to CMs is improved. shAJSZ improves cardiac function and reduces fibrosis, but this mechanism needs to be demonstrated.

Minor considerations:

Figures in the manuscript are not properly cited. For example, Fig S9 is not cited in the text.

Reviewer #2 (Remarks to the Author):

In their manuscript entitled " Conserved Transcription Factors Promote Cell Fate Stability and Restrict Reprogramming Potential in Differentiated Cells ", Alexandre R. Colas and colleagues reported four Transcription factors (ATF7IP, JUNB, SP7, and ZNF207) stabilize terminally differentiated cells state and act as barrier in TF-induced cell fate change in a lineage and cell type independent manner. Mechanically, AJSZ regulate chromatin accessibility dynamics and proximal transcription to promote cell fate stability and represses conserved network of genes required for reprogramming. The data presented in this manuscript looks having significant novelty. It will be suitable candidate for publication by revising the points listed below.

1 From Figure 1C and 1G, siAtf7ip seems more important than other three TFs for CR process. The TF ATF7IP shows more binding events and binding sites to closed chromatin than JUNB according Figure 3B and 3C. The author need explain why the number of JUNB binding sites of Domain2 is higher than ATF7IP? (Figure 4H)

2 whether AJSZ interact with each other directly at protein level?

3 How about the AJSZ gene expression of the cell population in the Cluster 2? Could it be possible that they diverged from the remaining cell population because of lower AJSZ expression?

Reviewer #3 (Remarks to the Author):

Induced pluripotent stem cells (iPSCs) are reprogrammed from differentiated cells and similarly, by using different set of transcription factors (TFs) or small molecules, one differentiated cell type such as fibroblast can be reprogrammed into another cell type such as cardiomyocyte (iCM) and neuron (iN). Both iPSCs and other induced cells are excellent cell source for regenerative medicine. However, up to now, the reprogramming efficiency is still very low and the cells generated are heterogenous, which greatly restrict their application in clinic. Although some reprogramming barriers have been identified, they were mainly validated in single reprogramming system. In this manuscript, Missinate et al, performed genome wide TF siRNA screen in cardiac reprogramming and identified 4 TFs AJSZ as reprogramming barriers. Moreover, the role of AJSZ is conserved across species, cell types and lineages. These findings are interesting and may be important for the improvement of reprogramming platform and generate high quality cells for therapeutic usage. However, this manuscript needs minor revision before considering for publication.

1. Though *in vivo* AJSZ KD with MGT overexpression improved cardiac function after myocardial infarction, no data showed the efficiency of in vivo reprogramming. The authors only used Zfp207 immunofluorescence to represent KD efficiency (Supp Fig 8 A, B). In Supp Fig 7A, Zfp207 was

abundantly expressed in CMs, what were the cells in Suppl Fig 8 B, CMs or fibroblast markers co-stain with Zfp207 should be performed. What is effect of Atf7ip and Zfp207 KD in CMs and these CMs should not be included in reprogramming efficiency assay. Instead of analyzing the *in vivo* AJSZ shRNA infection efficiency, they used the iMGT-MEF assay *in vitro* (Suppl Fig 7 B, C).

2. The authors demonstrated that AJSZ KD increased cardiac and neural reprogramming efficiency. How about the endoderm lineage such as hepatocyte and pancreatic cells?
3. The authors showed that AJSZ bound to open and close chromatin in ground state HDFs. What are the genes bound by ATF7IP, JUNB and ZNF207 in open chromatin? Are these genes specific for HDFs or overlapping with those bound by these 3 TFs in HAECs?
4. The authors concluded that JUNB plays a direct role in the establishment or maintenance of domain 1 in an open state and domain 2 in a closed state, also 97% of core promoters were bound by JUNB in HDFs. One might expect JUNB is the major player in opposing reprogramming. However, in Fig 1 C, siJunb did not increase Myh6-eGFP+ cells much, the most significant one is siAtf7ip, could the authors discuss about this?
5. In scATAC-seq, the authors identified cluster 2 as the cells undergoing reprogramming. It's about 13% of total cells which was similar to the 16% ACTN1+iCMs. What's the transfection efficiency of siAJSZ and GMT in this assay?
6. The authors discovered that ATF7IP and SP7 bound to closed chromatin regions enriched for SMAD and STAT motifs. Though they discussed their findings and the previously reported roles of TGF β /SMAD and STAT signaling pathways in reprogramming. It's still not clear the functions of SMAD and STAT in their reprogramming setting. Have they checked these two pathways after AJSZ KD and what is the effect of inhibition of these two pathways on AJSZ KD mediated reprogramming?
7. In the last section of Results, "Suppl Fig 7 C, D" should be Supp Fig 8 C, D. In Discussion section, "Supplementary Fig. 8A, 8B" should be Supplementary Fig 9A, 9B.
8. Some images lack scale bars or notes in figure legends and the English needs revision.

Response to Reviewers

I would like to thank the reviewers for the time and effort that has gone into reviewing and improving the paper. Overall, the reviews were positive and enthusiastic about the novelty of this manuscript. Please find our response to the comments in blue.

Reviewer #1 (Remarks to the Author):

Cell fate reprogramming is expected to be a new regenerative medicine, but the low reprogramming efficiency is an issue that needs to be solved. For successful cell fate conversion, the reprogramming factor must bind to the target DNA efficiently through epigenetic regulation or other means to regulate gene expression in the target cell type. However, there has been no systematic evaluation of factors as fate stabilizers in initiating cells. In this paper, to discover novel fate stabilizing regulators, the authors performed a genome-wide TF siRNA screen by applying the cardiac reprogramming assay in mouse embryonic fibroblasts and found that four transcription factors (ATF7IP, JUNB, SP7, and ZNF207 [AJSZ]) robustly inhibited cardiac reprogramming. Furthermore, the authors showed that these four factors are also common factors controlling fate stabilization when reprogramming primary human fibroblasts and adult endothelial cells into cardiomyocytes, neurons, and iPSCs.

Mechanistically, analysis using RNA-seq, ChIP-seq, and ATAC-seq revealed that AJSZ binds to both open and closed chromatin genome-wide and locally, limiting access to target DNA for reprogramming factors and chromatin remodeling, as well as by downregulating a series of conserved genes involved in the regulation of cell fate specification, proteome remodeling, ATP homeostasis, and inflammatory signaling, AJSZ limits the ability of cells to undergo large-scale phenotypic changes. Finally, the authors showed that even in vivo, the combination of KD of AJSZ and cardiac reprogramming can promote improvement in cardiac function and a decrease in fibrosis.

The concept is of interest as developing a new therapeutic strategy for reprogramming. However, I don't think the data is conclusive enough to draw a clear conclusion that the knockdown of the four factors robustly promotes reprogramming. They assessed reprogramming efficiency only by the percentage of specific markers but did not evaluate the function of the reprogrammed cells. Furthermore, quantitative evaluation of reprogramming efficiency should be done more precisely. It is also necessary to verify whether cells converted from AJSZ KD cells are equivalent to cells converted by conventional methods. The mechanism of the improvement of cardiac reprogramming in vivo has not been clarified, and more validation is needed.

My comments are as follows:

Major considerations:

1. First, the role of AJSZ needs to be clarified. The authors state that "AJSZ is expressed in most adult tissues but is down-regulated in lipolytic cells from different lines of cancer," but AJSZ was identified in a screen of immature mouse fetal fibroblasts (MEFs). It needs to be clarified whether these 4Fs are expressed in MEFs.

Good point. A new supplementary figure (**Supplementary Fig.2**) has been added to address this comment and shows basal expression of AJSZ in MEFs. Note that KD efficiency is also assessed in this context. This new data provides a comprehensive assessment of AJSZ protein levels at both baseline and upon KD in MEFs.

2. Since the cell cycle and reprogramming efficiency are strongly related, and the effect of siAtf7ip, which has been pointed out to be related to the cell cycle, is the largest among AJSZ (Fig 1C), it is necessary to clarify the effect of AJSZ KD on the cell cycle.

-Good point. Note that an effect of AJSZ KD on cell cycle genes is observed and reported in **Supplementary Table 12** (GO terms section) and **Fig.5c** (enrichment for the term: “Regulation of fibroblast proliferation”).

-In addition, we have generated new data to further address this comment and show that EDU incorporation (2 hours incubation) is significantly reduced in HDFs treated with MGT+siAJSZ and MGT+siATF7IP as compared to MGT+ siControl (**panel a and b below**). In addition, and consistent with these observations, our RNA-seq data reveal that a broad range of cell cycle regulators including genes involved in cell cycle progression (*CDK1*, *CDK2*, *BUB1*), DNA synthesis (*MKI67*) and cytokinesis (*ANLN*) are consistently downregulated in both MGT+siAJSZ and MGT+siATF7IP (**panel c below**) as compared to MGT+siControl. Thus, collectively these observations suggest that AJSZ KD via loss of ATF7IP function, reduce cell cycle activity and possibly promote cell cycle exit. Note that these data are consistent with a recent study (Bektik et al 2018, Int J Mol Sci) showing that iCMs exit cell cycle along the process of reprogramming with decreased percentage of 5-ethynyl-20-deoxyuridine (EdU)⁺. Thus, we propose that the reduced cell cycle activity observed upon AJSZ and ATF7IP KD, reflects the increase in reprogramming efficiency caused by siAJSZ and siATF7IP.

3. Although AJSZ is expressed in mature tissues, AJSZ KD promotes both reprogramming to undifferentiated and differentiated cells. Does AJSZ, which was KD before reprogramming, regain its expression in mature tissues after reprogramming, while it remains lost in undifferentiated cells?

-Good point, we have added new data showing the kinetic of expression recovery post-KD for the top 3 barriers (see **Supp Fig. 7g and below**). Note that by 6-day post-transfection (from Day-3 to Day 3), AJSZ regain 40-85% of control expression (**see arrow at day 3 below**). These results show that expression of barriers is regained over time post-KD as cells undergo reprogramming (day 3) and also confirm that barrier genes are actively transcribed in fibroblasts.

-Note these observations are consistent known effect of siRNAs on target genes: “Gene silencing resulting from siRNA can be assessed as early as 24 hours post-transfection. The effect most often will last from **5–7 days**.” (<https://www.sigmaaldrich.com/US/en/technical-documents/technical-article/genomics/gene-expression-and-silencing/sirna-faq>).

4. Although AJSZ KD shows an increase in Myh6-GFP positive cells, accurate quantitative evaluation by IHC is difficult; more accurate quantitative evaluation such as FACS would support the authors' results. The same can be said for other related reprogramming efficiency assessments.

-Image-based fluorescence quantification is a well-established technique to measure fluorescence in cells and has been used in multiple publications from our lab (*Colas et al 2012 Genes and Dev*; *Cunningham et al 2017 Genes and Dev*; *Elmen et al 2020 Disease Models and Mechanisms*; *Theis et al 2020, Elife*) and other labs (*Eulalio et al 2012 Nature*; *Wei et al 2015 Nature*; *Diaz-Cunado et al 2018 Cell Reports*). For this work, we have used an automated and high-throughput microscope (ImageXPress, Molecular Devices) located in the Colas lab and analyzed fluorescence signal using MetaXPress Analysis software suite (see methods).

-Also note that for all reprogramming experiments, a quantification of reprogramming efficiency was also performed by qPCR to confirm all image-based quantification of reprogramming efficiency.

-Finally, for reference, we have now performed a side-by-side comparison of both quantification methods to assess effect of AJSZ KD on reprogramming efficiency in the iMGT-MEF assay (see below). Note that for this experiment, we used a different iMGT MEFs clone, that has a higher reprogramming efficiency at baseline (~20%) than siControl in Fig.1 (~6%).

For the siAJSZ condition, both methods are in agreement and show that AJSZ KD induces ~80% of cells to reprogram in the iMGT-MEFs assay (see **panel a and b below**). However, for MGT+siControl condition, image-based fluorescence quantification measures that ~18% of cells express GFP, while the FACS method quantifies that 44% of cells express GFP (see **panel a and b below**). Collectively, these results highlight that FACS captures the whole continuum of fluorescence from very low to high intensities, while image-based fluorescence quantification mainly captures moderate to high fluorescence intensity values. Note that, regardless of the method used to quantify fluorescence, both techniques show that siAJSZ very significantly (p -value <0.0001) enhances MGT's ability to promote cardiac reprogramming. Thus, we conclude that both methods are equivalent to quantify moderate to high fluorescence intensity values, which is more relevant for reprogramming efficiency measurement, while FACS is more accurate to quantify low intensity values.

5. Is AJSZ KD relevant for reprogramming to more functionally mature cells? I mean is AJSZ KD relevant for full reprogramming. For example, does KD alter the efficiency of beating in cardiomyocytes, the induction of cardiomyocytes with clear sarcomeric structures, or the pluripotency in iPS cells?

Yes, the appearance of sarcomeric structures and calcium handling activity is unique to siAJSZ + MGT condition in HDFs (see **Supplementary Fig. 4c,d**), and thus suggest that AJSZ KD is required for the acquisition of these cardiac phenotypes.

-Also, we have generated new data showing that AJSZ KD does not alter hiPSC-derived CMs ability to handle calcium and beating frequency (see **Supp Fig. 10a-e and below**). Note number of cell analyzed per condition (n=787 for siControl and n= 665 for siAJSZ conditions)

6. Figure 2A shows that AJSZ is expressed in HDFs, but fibroblasts are a heterogeneous population. It is necessary to quantify the fibroblasts expressing each factor and to quantitatively evaluate the fibroblasts expressing AJSZ. The same can be said for related assessments in HAECs (Supplementary Fig. 4).

-Good point, we have added new data showing a wide field of view, highlighting that all fibroblasts express AJSZ. (see example below depicted in **Supplementary Figure 3c,d**).

-Regarding the heterogeneity of expression, among the 4 barrier TFs, JUNB shows the most variable expression pattern (**see high and low expressing cells below**), however, JUNB remains expressed in all cells albeit at different levels. Note that in this context, ATF7IP and ZNF207 show much less variability in expression. The functional significance of such variation in expression is likely to have an impact on the reprogramming competence of the cells, although this might be out of scope of the current study.

7. Supplementary Fig. 2A confirms the effect of siAJSZ, but HDFs should be added as a control; it should be shown whether MGT overexpression is independent of AJSZ expression. The same can be said for related assessments in HAECs (Supplementary Fig. 4).

Good point. Bar graphs have been updated with HDFs or HAECs as controls

8. Supplementary Fig. 2B shows that siAJSZ upregulates heart-related gene expression, but is there any change in fibroblast-related genes? It would be better if HDFs were necessary for control. The same can be said for other related assessments (Supplementary Fig. 3).

-Good point. Please refer to **Fig.5a**. We observe that fibroblast-related genes (i.e. *TAGLN*, *S100A4*, *GREM2*, *ADAMTS1*) are downregulated in response to AJSZ KD by day 3. Also, see below independent immunofluorescence picture showing loss of fibroblast gene expression (i.e. *TAGLN*) in iCMs.

9. Regarding Figures 2C and 2I, it is difficult to compare the expression of fibroblast markers/endothelial markers and CM markers only with the merged pictures. It is easier to understand if each picture is shown. Moreover, both of them seem to co-express CM markers and fibroblast or endothelial cell markers.

-Good point. Unmerged versions of the pictures for both fibroblast and endothelial cell have been added to **Supplementary Fig.4d** and **6e** respectively.

Can we confirm that the induced cells are cardiomyocytes?

- Yes, we observe phenotypic manifestations in reprogrammed cells that are consistent with a cardiac phenotype (i.e. sarcomeric structures and calcium handling), in addition to increased cardiac-specific gene expression (Fig.5a).

c**d**
10. Supplementary Figure 5 appears to contradict the authors' hypothesis that AJSZ is a general cell fate stabilizer. First, the comparison between Figure 5B and Figure 5C is confusing and should be integrated. Also, the reviewers believe that FACS is more accurate than IHC for quantitative evaluation. More critically, if AJSZ acts as a general cell fate stabilizer, it is hard to understand why the reprogramming efficiency decreases with time after AJSZ KD. This discrepancy needs to be verified.

-We disagree, from **Supplementary Fig.7 (old Supp Fig.5)**, our initial interpretation was that AJSZ expression is actively maintained in fibroblasts which is supported by the immunofluorescence data (**Fig. 2a**), thus if AJSZ are knocked down 4-5 days prior to the peak of MGT overexpression, then cells might be able to re-transcribe AJSZ and partially recover barrier function, which would explain why reprogramming efficiency is reduced when AJSZ is knocked down 3 days prior to Dox treatment.

-We have added new data and profiled expression of MGT and top 3 barrier TFs from day 0 to day 3. In this context, we observe that upon Dox treatment, MGT expression peaks at day 1 or 2 and that during this time frame, transfection of siAJSZ at day-1 condition elicits a greater reduction of top 3 barrier expression as compared to siAJSZ (day-3) (**Supplementary Fig. 7f,g**), which in turn correlates with increased reprogramming efficiency (**Supplementary Fig. 7d,e**). Thus, our interpretation is the following: cells' ability to undergo reprogramming is inversely proportional to barrier TF expression levels. Note that these findings further confirm the role of AJSZ as central regulators of fate stability in differentiated cells.

11. The results in Figure 3 show that JUNB and ATF7IP account for most of the binding events. In particular, JUNB not only recruits other factors and associates with open chromatin in the AP-1 motif enriched chromatin region but is also involved in maintaining the closed state of lineage-specific factors and seems to be the most important factor. However, Fig. 1C shows that siATF7IP promotes reprogramming efficiency the most. Please explain the reason for this.

-Good point, **new data (Supplementary Fig. 14)** and a **new paragraph in the discussion** have been added to address this comment. Briefly, new data shows that si*Aff7ip* causes a 20-50% reduction of *Junb*, *Sp7* and *Zfp207* protein levels, while alternatively si*Junb* only causes a reduction of *Junb* protein levels (**Supplementary Fig. 14a,b**). These results suggest that *Aff7ip* orchestrates *Junb*, *Sp7* and *Zfp207* barrier activity by regulating their mRNA expression or protein levels. Note that these findings are consistent with previous work (*Timms et al 2016 Cell reports*, *Tsusaka et al 2019 EMBO reports*), showing that *Atf7ip* can stabilize protein levels, by shielding interacting partners (i.e *Setdb1*) from proteasomal degradation. This mechanism is also consistent with newly added co-immunoprecipitation data showing that ATF7IP and ZNF207 directly interact (see **supplementary Fig.8a**).

-In conclusion, we propose that *Atf7ip* acts as an upstream regulator of the barrier network, by directly or indirectly stabilizing *Junb*, *Sp7* and *Zfp207* protein levels, thereby concentrating most of the barrier activity. In this model, *Junb* plays a fraction of the role orchestrated by *Atf7ip*, as a downstream effector (see **Supplementary Fig.13c**).

12. In fig4B, on t-sne, Cluster2 is a new cluster emerged by AJSZ KD, but does this cluster still exist as a discrete cell population in the integrated dataset of MGT+siCtrl and MGT+siAJSZ?

-We merged the MGT+siCtrl and MGT+siAJSZ scATAC-seq data using cellranger's aggr function. In this case, clustering of these samples using both t-SNE and UMAP did not visually separate any group of cells. It is not clear why the integrated dataset did not show a separate cluster. We speculate that it did not cluster out due to the noise from combining multiple datasets. In addition, in vitro early timepoints generally show less distinct cell patterns, especially at such an early point following reprogramming.

13. Supplementary Fig. 6B shows Nkx2.5 as a representative of differentially accessible (DA) transcriptional start sites. However, it seems that cardiac reprogramming converts fibroblasts directly into cardiomyocytes without going through cardiac progenitor cells expressing Nkx2.5. Why the change in cardiac stem cell markers, and is it possible that AJSZ KD reprogrammed cardiac stem cells?

-NKX2-5 is expressed both in cardiac progenitors and differentiated CMs (see human protein atlas), therefore NKX2-5 is not specific to either state. However, consistent with our interpretation, TSS of additional cardiomyocyte markers such as NPPA, MYOM1, ACTA1, MYH7 (see **Supplementary Table 9**) are also differentially open in cluster 2. In contrast cardiac mesoderm markers such as MESP1 or EOMES are not differentially open. Based on these observations, we propose that NKX2-5 can be used as a CM and not as a progenitor marker, and therefore indicate that AJSZ KD enhances MGT-mediated direct cardiac reprogramming without transitioning through a progenitor stage as previously described in *Ieda et al 2010 Cell*.

14. Other groups have already shown that forced expression of reprogramming factors can change the state of chromatin. In Fig. 4D, Domain 1 is compared with Domain 2 as unchallenged HDFs and HDF, HDFs. Wouldn't it be more appropriate to compare the three groups, HDF, siControl+MGT, siAJSZ+MGT, to evaluate the target of AJSZ?

-Overexpression of MGT alone is not sufficient to induce efficient reprogramming in human fibroblasts, as previously described by other groups (*Nam et al 2013 PNAS*, *Zhou et al 2019 Cell Stem Cell*) and in this study. MGT overexpression alone is only able to induce significant chromatin remodeling *in mouse fibroblasts* (*Stone et al 2019 Cell Stem Cell*). Here, our goal was to identify a chromatin accessibility profile discriminating cells that reprogram from those that do not (i.e ground state or reprogramming resistant cells) and describe the molecular differences between both states. Remarkably, note that this strategy enabled to identify that differential accessibility of AP-1 or reprogramming TF (i.e MEF2C) motif-enriched regions correlates with the ability of cells to either resist or undergo reprogramming respectively.

15. In figure 7, EF after myocardial infarction seems to be too low. It seems to be around 20% in many studies. Was there any difference in survival rate among the three groups? Also, in figure 7E and F, MI + PBS should be shown to evaluate whether the myocardial infarction was properly prepared.

-All quantification related to heart function was performed by highly trained personnel at the Small Animal Cardiovascular Phenotyping and Model Core (Johns Hopkins University, Department of Medicine) in a blinded fashion.

-No survival difference was observed between the different groups (see methods)

-MI+PBS condition has been added to all histograms (see **Figure 7**)

16. Supp Fig.7A alone, it is difficult to show whether AJSZ is indeed expressed in non-cardiomyocytes (especially cardiac fibroblasts). Even if AJSZ is expressed in cardiac fibroblasts, it is necessary to mention the percentage of fibroblasts expressing AJSZ and the localization of the cells (remote, border, infarct area).

Good point, unfortunately, the antibody repertoire (Tnnt2, AJSZ, Col1) does not allow for co-staining as all the antibodies are raised in rabbit. However, note that cardiac and fibrotic (injured) areas are morphologically different and can be recognized. In addition, a new antibody staining the fibrotic compartment (Col1) has been added to **new Supplementary Fig.11c**. Thus, these morphological differences and newly added Col1 staining, enable to determine whether AJSZ expression is located in the cardiac and/or fibrotic compartment.

17. Supp Fig.7B,C also need to be quantitatively evaluated by FACS. The shControl in Fig. 7C appears to have almost no GFP+ cells.

In shControl condition, GFP+ cells appear smaller and dimer, which might contribute to the impression that the difference between shcontrol and shAJSZ is greater than 4.

For quantification strategy, see response to comment #3.

18. Supp Fig.8 also needs to show the exact KD efficiency of AJSZ with WB and qRT if possible. In addition, MI+PBS needs to be added as a control in 8A.

KD efficiency for shAJSZ is measured *in vivo* by quantifying the % of Zfp207 cells in MGT+ShControl vs MGT+shAJSZ conditions (see **Supplementary Fig12a,b**).

19. It is necessary to show whether the efficiency of *in vivo* reprogramming from fibroblasts to CMs is improved. shAJSZ improves cardiac function and reduces fibrosis, but this mechanism needs to be demonstrated.

-Great point. New data has been added in **Figure 7** to address this comment and shows that the scar size (Masson trichrome) and the % of Col1 expressing fibroblasts is reduced, while cardiac content is enhanced (% of Tnnt2+ cells) in sections from MGT+shAJSZ as compared to MGT or PBS conditions. Thus, given that 1) previous studies (*Qian et al 2012 Nature, Inagawa et al 2012 Circ. Res*) have already demonstrated that MGT overexpression induces cardiac reprogramming *in vivo* and improves heart function and 2) our data show that AJSZ KD enhances reprogramming efficiency *in vitro* and improves heart function *in vivo*, hence we propose that the likely mechanism by which heart function is improved in MGT+shAJSZ condition, is *via* enhanced reprogramming efficiency (Note that text mentioning this interpretation has been added in both results and discussion sections). A direct demonstration of the reprogramming mechanism *in vivo* requires the acquisition and breeding of mouse lines (i.e Rosa26-mcherry (R26R) and Postn-Cre- lines) that is not compatible with the timeline of the review process (>6 months).

-Finally, note that 5 new data points per condition have been added to strengthen *in vivo* conclusion study and improve the statistical significance of the findings (**Fig. 7i,j**).

Minor considerations:

Figures in the manuscript are not properly cited. For example, Fig S9 is not cited in the text.

Figure citations have been updated

Reviewer #2 (Remarks to the Author):

In their manuscript entitled " Conserved Transcription Factors Promote Cell Fate Stability and Restrict Reprogramming Potential in Differentiated Cells ", Alexandre R. Colas and colleagues reported four Transcription factors (ATF7IP, JUNB, SP7, and ZNF207) stabilize terminally differentiated cells state and act as barrier in TF-induced cell fate change in a lineage and cell type independent manner. Mechanically, AJSZ regulate chromatin accessibility dynamics and proximal transcription to promote cell fate stability and represses conserved network of genes required for reprogramming. The data presented in this manuscript looks having significant novelty. It will be suitable candidate for publication by revising the points listed below.

1 From Figure 1C and 1G, siAtf7ip seems more important than other three TFs for CR process. The TF ATF7IP shows more binding events and binding sites to closed chromatin than JUNB according to Figure 3B and 3C. The author need explain why the number of JUNB binding sites of Domain2 is higher than ATF7IP? (Figure 4H)

Good point, **new data (Supplementary Fig. 14)** and a **new paragraph in the discussion** have been added to address this comment. Briefly, new data shows that siAtf7ip causes a 20-50% reduction of Junb, Sp7 and Zfp207 protein levels, while alternatively siJunb only causes a reduction of Junb protein levels (**Supplementary Fig. 14a,b**). These results suggest that Atf7ip orchestrates Junb, Sp7 and Zfp207 barrier activity by regulating their expression/protein levels. Note that these findings are consistent with previous work (*Timms et al 2016 Cell reports, Tsusaka et al 2019 EMBO reports*), demonstrating that Atf7ip can stabilize protein levels, by shielding interacting partners (i.e Setdb1) from proteasomal degradation. This mechanism is also consistent with newly added co-immunoprecipitation data showing that ATF7IP and ZNF207 interact at the protein level (see **supplementary Fig.8a**).

In sum, we propose that Atf7ip acts as an upstream regulator of the barrier network, by directly or indirectly stabilizing Junb, Sp7 and Zfp207 protein levels, thereby concentrating most of the barrier activity. In this model, Junb plays a fraction of the role orchestrated by Atf7ip, as a downstream effector (see **Supplementary Fig.14c**).

2 whether AJSZ interact with each other directly at protein level?

Great point, we have added **new data** to address this comment. Briefly, pulldown of ZNF207 followed by immunoblotting, revealed that ZNF207 interacts with ATF7IP and JUNB (**Supplementary Fig.8a**). These findings are consistent with data showing chromatin co-occupancy for JZ, AZ and AJZ (see **Fig. 3e** and **Supplementary Table 5**) and that ATF7IP might regulate ZNF207 and other fate stabilizers (JUNB and SP7) protein levels via physical interaction (**Supplementary Fig14a-c**). **Fig.3 k** has been updated accordingly.

3 How about the AJSZ gene expression of the cell population in the Cluster 2? Could it be possible that they diverged from the remaining cell population because of lower AJSZ expression?

Great point, we have added **new data** showing that MGT transduction efficiency in HDFs is ~25-30% (**Supplementary Fig.3a,b**), thereby suggesting that cluster 2 (**Fig. 4b**) actually represents $\sim 13/30 = 43\%$ of transduced cells. This implies that based on single cell ATAC-seq (Fig.4b) and immunofluorescence (Fig.2c) data, $\sim 43\%$ of MGT-transduced cells undergo reprogramming in HDFs upon AJSZ KD. Note that this % is similar to the reprogramming efficiency observed in the iMGT-MEFs assay, where MGT+siAJSZ condition induces 36-40% of MEFs to reprogram (**Fig. 1h**).

Reviewer #3 (Remarks to the Author):

Induced pluripotent stem cells (iPSCs) are reprogrammed from differentiated cells and similarly, by using different set of transcription factors (TFs) or small molecules, one differentiated cell type such as fibroblast can be reprogrammed into another cell type such as cardiomyocyte (iCM) and neuron (iN). Both iPSCs and other induced cells are excellent cell source for regenerative medicine. However, up to now, the reprogramming efficiency is still very low and the cells generated are heterogenous, which greatly restrict their application in clinic. Although some reprogramming barriers have been identified, they were mainly validated in single reprogramming system. In this manuscript, Missinato et al, performed genome wide TF siRNA screen in cardiac

reprogramming and identified 4 TFs AJSZ as reprogramming barriers. Moreover, the role of AJSZ is conserved across species, cell types and lineages. These findings are interesting and may be important for the improvement of reprogramming platform and generate high quality cells for therapeutic usage. However, this manuscript needs minor revision before considering for publication.

1. Though *in vivo* AJSZ KD with MGT overexpression improved cardiac function after myocardial infarction, no data showed the efficiency of *in vivo* reprogramming.

-Great point. New data has been added in **Figure 7** to address this comment and shows that the scar size (% of Col1+ cells) is reduced while cardiac content is enhanced (% of Tnnt2+ cells) in sections from MGT+shAJSZ as compared to MGT or PBS conditions. Thus, given that 1) previous studies (*Qian et al 2012 Nature, Inagawa et al 2012 Circ. Res*) have already demonstrated that MGT overexpression induces cardiac reprogramming *in vivo* and improves heart function and 2) our data show that AJSZ KD enhances reprogramming efficiency *in vitro* and improves heart function *in vivo*, hence we propose that the likely mechanism by which heart function is improved in MGT+shAJSZ condition, is *via* enhanced reprogramming efficiency (Note that text mentioning this interpretation has been added in both results and discussion sections). A direct demonstration of the reprogramming mechanism *in vivo* requires the acquisition and breeding of mouse lines (i.e Rosa26-mcherry (R26R) and Postn-Cre- lines) that is not compatible with the timeline of the review process (>6 months).

-Finally, note that 5 new data points per condition have been added to strengthen *in vivo* conclusion study and improve the statistical significance of the findings (**Fig. 7i,j**).

The authors only used Zfp207 immunofluorescence to represent KD efficiency (Supp Fig 8 A, B). In Supp Fig 7A, Zfp207 was abundantly expressed in CMs, what were the cells in Suppl Fig 8 B, CMs or fibroblast markers co-stain with Zfp207 should be performed.

Good point, unfortunately, the antibody repertoire (Tnnt2, AJSZ, Col1a) does not allow for co-staining as all the antibodies are raised in rabbit. However, note that cardiac and fibrotic (injured) areas are morphologically different from each other and can be annotated based on these features. In addition, a new antibody staining the fibrotic compartment (Col1a) has been added to **new Supplementary Fig.11c** and confirms morphological assessment of the compartments. Thus, using these morphological differences and newly added Col1a staining, we are able to annotate whether AJSZ expression is located in the cardiac and/or fibrotic compartment.

What is effect of Atf7ip and Zfp207 KD in CMs and these CMs should not be included in reprogramming efficiency assay.

Good point. **New data (Supplementary Fig. 10)** has been added showing that AJSZ KD does not affect hiPSC-CMs beating frequency nor calcium handling.

Instead of analyzing the *in vivo* AJSZ shRNA infection efficiency, they used the iMGT-MEF assay *in vitro* (Suppl Fig 7 B, C).

KD efficiency for shAJSZ is measured *in vivo* by quantifying the % of Zfp207 cells in MGT+ShControl vs MGT+shAJSZ (see Supplementary Fig. 12a,b).

2. The authors demonstrated that AJSZ KD increased cardiac and neural reprogramming efficiency. How about the endoderm lineage such as hepatocyte and pancreatic cells?

Good point, we have not established a reprogramming assay for endoderm lineages but based on the findings that the barrier role of AJSZ is lineage independent, we predict the AJSZ should also play a barrier role during hepatocyte or pancreatic cells reprogramming. Note that a reference to this comment is made in the last paragraph of the discussion.

3. The authors showed that AJSZ bound to open and close chromatin in ground state HDFs. What are the genes bound by ATF7IP, JUNB and ZNF207 in open chromatin? Are these genes specific for HDFs or overlapping with those bound by these 3 TFs in HAECs?

-Great point **new data** has been added (see new **Fig.3f**, **new Supplementary Figure 8** and **new Table 6**), that describe genes bound by ATF7IP, JUNB and ZNF207 in open chromatin. Main text has been edited accordingly.

-We have not performed ChIP-seq in HAECs, however our expression data show a conserved role for AJSZ in the regulation of the 5 reprogramming agonists (please refer to **Figure 6 i,j**).

4. The authors concluded that JUNB plays a direct role in the establishment or maintenance of domain 1 in an open state and domain 2 in a closed state, also 97% of core promoters were bound by JUNB in HDFs. One might expect JUNB is the major player in opposing reprogramming. However, in Fig 1 C, siJunb did not increase Myh6-eGFP+ cells much, the most significant one is siAtf7ip, could the authors discuss about this?

Good point, **new data (Supplementary Fig. 14)** and a **new paragraph in the discussion** have been added to address this comment. Briefly, new data shows that siAtf7ip causes a 20-50% reduction of Junb, Sp7 and Zfp207 protein levels, while alternatively siJunb only causes a reduction of Junb protein levels (**Supplementary Fig. 14a,b**). These results suggest that Atf7ip orchestrates Junb, Sp7 and Zfp207 barrier activity by regulating their expression/protein levels. Note that these findings are consistent with previous work (*Timms et al 2016 Cell reports*, *Tsusaka et al 2019 EMBO reports*), demonstrating that Atf7ip can stabilize protein levels, by shielding interacting partners (i.e Setdb1) from proteasomal degradation. This mechanism is also consistent with newly added co-immunoprecipitation data showing that ATF7IP and ZNF207 interact at the protein level (see **supplementary Fig.8a**).

In sum, we propose that Atf7ip acts as an upstream regulator of the barrier network, by directly or indirectly stabilizing Junb, Sp7 and Zfp207 protein levels, thereby concentrating most of the barrier activity. In this model, Junb plays a fraction of the role orchestrated by Atf7ip, as a downstream effector (see **Supplementary Fig.14c**).

5. In scATAC-seq, the authors identified cluster 2 as the cells undergoing reprogramming. It's about 13% of total cells which was similar to the 16% ACTN1+iCMs. What's the transfection efficiency of siAJSZ and GMT in this assay?

Great point, we have added **new data** showing that MGT transduction efficiency in HDFs is ~25-30% (Supp Fig.3a,b), thereby suggesting that cluster 2 (Fig. 4b) actually represents ~13/30= 43% of transduced cells. This implies that based on single cell ATAC-seq (Fig.4b) and immunofluorescence (Fig.2c) data, ~43% of MGT-transduced cells undergo reprogramming in HDFs upon AJSZ KD. Note that this % is similar to the reprogramming efficiency observed in the iMGT-MEFs assay where MGT+siAJSZ condition induces 36-40% of MEFs to reprogram (Fig. 1h).

6. The authors discovered that ATF7IP and SP7 bound to closed chromatin regions enriched for SMAD and STAT motifs. Though they discussed their findings and the previously reported roles of TGF β /SMAD and STAT signaling pathways in reprogramming. It's still not clear the functions of SMAD and STAT in their reprogramming setting. Have they checked these two pathways after AJSZ KD and what is the effect of inhibition of these two pathways on AJSZ KD mediated reprogramming?

Great point, **new data** has been added (see Fig.3 h,i), main text and discussion have been updated accordingly. Briefly, new data shows that siStat4/5/6 increase reprogramming efficiency in the iMGT-MEF assay, suggesting that ATF7IP and SP7 may cooperate with STAT4/5/6 at discrete regions of closed chromatin to stabilize cell fate and oppose reprogramming.

Also, note that we have removed SMAD motif from the Fig.3g table, as most enriched motifs are STATs and siSmads did not increase reprogramming efficiency (data not shown).

7. In the last section of Results, “Suppl Fig 7 C, D” should be Supp Fig 8 C, D. In Discussion section, “Supplementary Fig. 8A, 8B” should be Supplementary Fig 9A, 9B.

Reference to Figures has now been updated

8. Some images lack scale bars or notes in figure legends and the English needs revision.

We have added missing scale bars.

REVIEWER COMMENTS

Reviewer #1 (Remarks to the Author):

It is still unclear whether KD of AJSZ promotes in vivo cardiac reprogramming or just reduces fibrosis without lineage tracing experiments in fig 7. If Atf7ip alone can inhibit cell proliferation as they have shown, it is difficult to distinguish whether the lentiviral AJSZ promoted reprogramming or inhibited fibroblast proliferation and fibrosis in vivo. Since CFs were not labeled immunohistochemically in FigS12b, the immuno+ cells appear to be CMs morphologically. Actn1 should be Actn2 for ICC.

Reviewer #2 (Remarks to the Author):

In the revised manuscript, the authors seriously performed reasonable experiments and provided reliable data for the major concerns mentioned before. I sincerely hope this manuscript can be published on Nature Communications as this work will have a profound impact on reprogramming field.

Reviewer #3 (Remarks to the Author):

The authors have addressed the reviewers' comments properly and substantially revised their manuscript. There is a typo in the combined PDF file, page 10 'siControl- or siAJSZ- transfected HDFs using we scATAC-seq'. The word 'we' should be deleted. This manuscript is now suitable for publication in Nature Communications.

Response to Reviewers

I would like to thank the editor and the reviewers for the time and effort that has gone into reviewing and improving the paper. Overall, the reviews from the previous round of revision were quite positive as reviewers #2 and 3 were enthusiastic about the novelty and impact of the study on the reprogramming field and thought that the revised manuscript was suitable for publication in Nature Communications. Reviewer #1, however, had raised one last concern regarding data supporting the efficacy of in vivo knockdown and whether fibroblasts were transduced. In this context, the editor (correspondence from May 11th) had asked the authors to provide significant qualitative and quantitative data, along with multiple representative images to support the in vivo claims.

Response to reviewer#1

Approach. To demonstrate efficacy of *in vivo* KD, mice were subjected to MI as in Figure 7. Next, three days post-MI, treatments ((PBS (= control) or MGT+shAJSZ)) were delivered at the site of injury by ultrasound-guided injection (see methods). 4 days post-injection, hearts were harvested, cryopreserved and serially sectioned to enable antibody staining on consecutive sections as all antibodies used here are raised in rabbit which precludes co-staining. **In this context, AJSZ KD efficiency is quantified by measuring the percentage of Atf7ip and Zfp207 + cells in MGT+shAJSZ condition as compared to PBS (see methods section below).** Note that among the antibodies directed against the four barrier TFs, Atf7ip and Zfp207 generate the highest signal to noise ratio on sections and are therefore best suited for image-based fluorescence quantification. Also note that in this context, we expect gene KD to be more pronounced at the site of injury as compared to uninjured regions, as injections were performed at the site of injury.

Results. To determine the area of injury, we first performed immunostaining for Col1a in both PBS and MGT+shAJSZ conditions (**panel a,b**). Note that the area of Col1a+ expressing cells was similar for both conditions (see white outline in **Panel a**). In this context, the area of Col1a staining was used to outline the injured regions in Atf7ip and Zfp207 staining images (see solid white line outline in **panel c and g**).

Next, to evaluate the efficacy of KD, we performed Zfp207 staining on consecutive serial sections (**panel c,d**).

In this context, we could observe large areas of Zfp207 KD (dotted white line highlights region with ~80% KD, see **panel e,f**) in the injured region of MGT+shAJSZ condition, while in contrast, Zfp207 expression was not affected in injured and non-injured areas of PBS condition. These phenotypes could be observed across multiple sections (n=3) and hearts (n=2).

Consistent with this observation, immunostaining for Atf7ip also revealed the presence of large areas of Atf7ip reduced expression in the injured region of the heart in MGT+shAJSZ condition as compared to PBS (**panel g-j**). Collectively, these results show that the echo-assisted injection

of multiple lentiviruses targeting the barrier TFs represents an efficient method to induce AJSZ KD in large areas of the injured heart. In turn, these large areas of KD are consistent the significant

improvement in heart function observed in MGT+shAJSZ treatment as compared to PBS or MGT treatments (see **Figure 7** of the manuscript).

Methods: Each section presented here, is a composite of ~60-100 images acquired at 20X with ImageXpress microscope (Molecular Devices). To generate the composite image, all 20X images, for each section, were stitched together. Finally, antibody signal intensity was quantified using a image analysis software (MetaXpress, Molecular Devices as described in methods of the main manuscript). Examples of masks generated by the software Zfp207 and DAPI are shown below.

- Both “shAJSZ enhance MGT-mediated improvement of heart function after myocardial infarction” paragraph and the discussion have been modified to incorporate these new results (see edits in blue).

REVIEWERS' COMMENTS

Reviewer #1 (Remarks to the Author):

Missinato and colleagues have substantially revised their manuscript and provide compelling evidence that Zfp207 and Atf7ip are effectively knocked down at the infarct zone four days post-MI. However, it remains to be proven that the cells knocked down by lentiviral AJSZ are indeed exclusively cardiac fibroblasts. In the new Fig. S12a,b, which shows the heart at low magnification, it is difficult to demonstrate whether transduction occurred only in the fibroblasts. The infarcted zone several days after MI, which the authors analyzed, is not only comprised of proliferating fibroblasts, but also impaired cardiomyocytes. Additionally, a significant number of inflammatory cells, which are closely associated with fibrosis and impaired cardiac function after MI, have infiltrated the infarcted area, and lentiviral vectors are capable of gene transfer to these cells. The reviewer expresses concern that the in vivo results from AJSZ knockdown may truly be a fibroblast-specific effect or an overestimation due to the effects on multiple cardiac cell types.

Response to Reviewers

I would like to thank the editorial team for accepting, in principle, our study for publication at Nature Communications and the reviewers for the time and effort that has gone into reviewing and improving the paper. In the last round of revision, Reviewer#1 has assessed that the authors have substantially revised their manuscript and provide compelling evidence that Zfp207 and Atf7ip are effectively knocked down at the infarct zone four days post-MI has one last comment, which was the focus of his/her last comment: In this context, reviewr#1 has one last remark that is addressed below in blue.

Reviewer #1

Missinato and colleagues have substantially revised their manuscript and provide compelling evidence that Zfp207 and Atf7ip are effectively knocked down at the infarct zone four days post-MI. However, it remains to be proven that the cells knocked down by lentiviral AJSZ are indeed exclusively cardiac fibroblasts. In the new Fig. S12a,b, which shows the heart at low magnification, it is difficult to demonstrate whether transduction occurred only in the fibroblasts. The infarcted zone several days after MI, which the authors analyzed, is not only comprised of proliferating fibroblasts, but also impaired cardiomyocytes. Additionally, a significant number of inflammatory cells, which are closely associated with fibrosis and impaired cardiac function after MI, have infiltrated the infarcted area, and lentiviral vectors are capable of gene transfer to these cells. The reviewer expresses concern that the in vivo results from AJSZ knockdown may truly be a fibroblast-specific effect or an overestimation due to the effects on multiple cardiac cell types.

Response to reviewer#1

We agree with reviewer #1 and are not claiming that AJSZ KD solely occurs in fibroblasts, but rather at site of injury marked by Col1 staining. Note that, KD is most prominent at this region because the echo-assisted injection was targeting the non-contracting part of the heart (injured area). We agree that the injured region of the heart is composed of a mixture of cells (myofibroblasts, immune cells and some cardiomyocytes), that can all be transduced by lentiviruses. Note that, our data from **Figure 1** and **2**, show that loss of AJSZ function enhances cardiac reprogramming in a cell type-independent manner. Based on these observations, we concur with reviewer#1 that cardiac reprogramming can occur in fibroblasts and other cell types to contribute to this process and promote heart repair. To incorporate this remark in the manuscript, we now mention in the last paragraph of the discussion that “more work is needed to describe the cell types and cellular mechanisms mediating this improvement”, which acknowledges reviewer#1’s remark.